# Transcriptional read-through of the long non-coding RNA *SVALKA* governs plant cold acclimation

Peter Kindgren [1], Ryan Ard [1], Maxim Ivanov [1] & Sebastian Marquardt [1]

Most DNA in the genomes of higher organisms does not encode proteins, yet much is transcribed by RNA polymerase II (RNAPII) into long non-coding RNAs (lncRNAs). The biological significance of most lncRNAs is largely unclear. Here, we identify a lncRNA (*SVALKA*) in a cold-sensitive region of the *Arabidopsis* genome. Mutations in *SVALKA* affect *CBF1* expression and freezing tolerance. RNAPII read-through transcription of *SVALKA* results in a cryptic lncRNA overlapping *CBF1* on the antisense strand, termed *asCBF1*. Our molecular dissection reveals that *CBF1* is suppressed by RNAPII collision stemming from the *SVALKA-asCBF1* lncRNA cascade. The *SVALKA-asCBF1* cascade provides a mechanism to tightly control *CBF1* expression and timing that could be exploited to maximize freezing tolerance with mitigated fitness costs. Our results provide a compelling example of local gene regulation by lncRNA transcription having a profound impact on the ability of plants to appropriately acclimate to challenging environmental conditions.

[1] Department of Plant and Environmental Sciences, Copenhagen Plant Science Centre, University of Copenhagen, Bulowsvej 34, Frederiksberg 1871, Denmark. Correspondence and requests for materials should be addressed to S.M. (email: sebastian.marquardt@plen.ku.dk)

RNA Polymerase II (RNAPII) transcription in genomes results in the production of many long non-coding RNAs (lncRNAs)[1]. The functional significance of resulting lncRNA molecules is actively debated even though biological roles have been identified for an increasing number of examples[2–4]. Expression of lncRNAs is remarkably specific to the environmental condition, tissue or cell type, arguing for roles of lncRNAs in regulation[5–7]. In addition to functions carried out by lncRNA molecules, the process of transcribing non-coding DNA sequences can by itself be regulatory in many systems[8,9]. Non-coding DNA regions in genomes can therefore affect gene expression by different mechanisms that need to be resolved experimentally.

Sessile organisms respond to changing environmental conditions by the regulation of gene expression. Early events in the *Arabidopsis* cold response include rapid transcriptional up-regulation of the intron-less C-repeat/dehydration-responsive element binding factors (CBFs)[10,11]. The CBFs are highly conserved transcription factors that promote cold tolerance in many plant species and are often arranged in a single cluster[12]. *CBF* expression during cold exposure activates downstream *COLD REGULATED (COR)* genes that promote freezing tolerance by adjusting the physiological and biochemical properties of plant cell interiors[13–15]. Intriguingly, constitutive *CBF* expression increases cold tolerance but is also associated with fitness penalties[16–19]. For example, over-expression of *CBF1* results in increased tolerance to freezing temperatures, but also severely reduces biomass[20]. Therefore, the expression of the endogenous *CBF1* gene is characterized by a transient peak of maximal expression during cold acclimation[11]. The negative fitness costs of both increased and decreased *CBF* expression highlight the importance of their tight regulation, yet the specific mechanisms that tightly control *CBF* expression during the early response to cold are not fully understood.

The plant response following long exposure to cold temperatures is associated with numerous lncRNAs that associate with the *FLOWERING LOCUS C (FLC)* gene. The antisense lncRNA *COOLAIR* is induced after 2 weeks of cold, and initiates chromatin repression of the *FLC* gene[2,21]. Full *FLC* silencing is aided by polycomb repressive complex recruitment by the intron-derived lncRNA *COLDAIR* and reinforced by the promoter derived lncRNA *COLDWRAP*[22]. This cold-triggered cascade of lncRNAs establishes long-lasting and stable repression of the *FLC* gene and serves as a paradigm for gene repression by lncRNAs[23]. However, it is currently unresolved if early responses to cold temperatures in plants are mediated by lncRNAs.

Here, we map transcriptional start sites in *Arabidopsis* during early responses to cold temperatures. We identify a cascade of two lncRNAs that fine-tune the expression of the *CBF1* gene. Transcriptional read-through of the lncRNA *SVALKA* results in the expression of a cryptic antisense *CBF1* lncRNA (*asCBF1*). *asCBF1* transcription results in RNAPII collision to limit the expression of full-length *CBF1*. This work extends the biological roles of lncRNAs during cold acclimation and it provides an elegant mechanism that achieves rapid dynamic regulation of environmentally sensitive gene expression.

## Results

### Identification of the lncRNA SVALKA. To identify transcription initiation events that respond to cold temperature in *Arabidopsis*, we performed transcription start site (TSS)-sequencing(TSS-seq)[24,25]. We obtained 184 million raw reads from two biological replicates at 22 °C and two biological replicates at 3 h of 4 °C. From these, we called 14250 TSS clusters. Our analyses revealed 489 down-regulated and 1404 up-regulated TSSs in response to 3 h at 4 °C (Fig. 1a, Supplementary Data 1, Supplementary Data 2). Most

differentially used TSSs were located in promoter regions corresponding to changed promoter usage in fluctuating environments. Of the up-regulated genes (±200 bp of the major TSS), the *CBF* genes were upregulated 100–400 fold making the *CBF* genomic region by far the most cold-responsive region in the genome (Fig. 1b). Interestingly, we detected a cold-responsive lncRNA, transcribed on the antisense strand between *CBF3* and *CBF1* (Fig. 1c) that we named *SVALKA* (*SVK*, for short). We found two TSSs by Rapid Amplification of cDNA Ends (RACE) that corresponded to the peaks observed with TSS-seq (Fig. 1d). Our 3′RACE analysis also supported the existence of alternative *SVK* isoforms. Notably, we detected no transcript overlapping the downstream *CBF1-3′UTR* (Supplementary Figure 1, Supplementary Figure 2). RT-qPCR revealed that more *SVK* is generated from the *CBF1*-proximal promoter than from the distal promoter (Supplementary Figure 3), consistent with TSS-seq analyses. A time series of cold exposure (4 °C) with samples taken every 4 h revealed the expression pattern of *SVK* and *CBF1* (Fig. 1e). *CBF1* expression peaked at 4 h followed by a decrease that reached a relatively stable level after 12–16 h, a pattern in congruence with earlier studies[26]. In contrast, *SVK* showed gradual increase in expression from 4 h to reach a stable maximum level after 12–16 h.

### SVK represses CBF1 and promotes cold acclimation. The anti-correlation between *CBF1* and *SVK* and their genomic proximity suggested that *SVK* could be involved in *CBF1* repression. To test this, we constructed two distinct *LUCIFERASE (LUC)* reporter lines of *CBF1* with different termination sequences. We expected any repressive role of *SVK* to be present in the lines with endogenous 3′-sequences (*CBF1:SVK*). In contrast, we expected effects of *SVK* to be absent in the lines with the terminator sequences of the *NOPALINE SYNTHASE (NOS)* gene that shows low antisense transcription[4] (*CBF1:T_{NOS}*). We subjected three independent lines from each construct to 4 °C for 0–12 h and subsequently measured LUC activity (Fig. 2a, b). The three *CBF1:T_{NOS}* lines showed increased *LUC* activity compared to the *CBF1:SVK* lines. These data demonstrated that *SVK* represses *LUC* activity. We wanted to confirm these findings by measuring antisense transcription from the 3′-end of the *LUC* constructs. Therefore, we designed a qPCR based strategy to measure antisense transcription that may be initiated from the endogenous CBF1–3′-UTR or the $T_{NOS}$ terminator[4]. We detected antisense transcription from the $T_{NOS}$ sequence in two independent *CBF1:T_{NOS}* lines (Supplementary Figure 4). However, antisense transcription from $T_{NOS}$ was not cold-responsive, arguing against a cold-induced effect of *LUC* expression. In contrast, we detected cold-induced antisense akin to *SVALKA* transcription from the endogenous *CBF1−3′*-UTR in the *CBF1:SVK* lines (Supplementary Figure 4). These data suggest that cold-induced antisense transcription could be involved in regulating endogenous *CBF1* expression.

We next characterized *Arabidopsis* Transfer DNA (T-DNA) lines to further dissect the function of the *CBF1* 3′-region. T-DNA lines insert large sequences of DNA (often 4–5 kb) in the *Arabidopsis* genome that disrupt the genomic context at the insertion site[27]. We isolated three informative T-DNA insertion mutants: (1) a line that disrupted *SVK* (*svk-1*), (2) a line that increased the distance of *SVK* transcription from *CBF1* (*uncoupling svalka-1, uns-1*), and (3) a line that showed *SVK* over-expression (*SVK OE*) (Fig. 2c, Supplementary Figure 2, 5). The *SVK* over-expression originates from strong promoters within the T-DNA that continued over the insertion border (Supplementary Figure 5a-b). We could not detect any other T-DNA driven transcription in the *svk-1* or *uns-1* mutants (Supplementary Figure 5c-d). We quantified *SVK* and *CBF1*

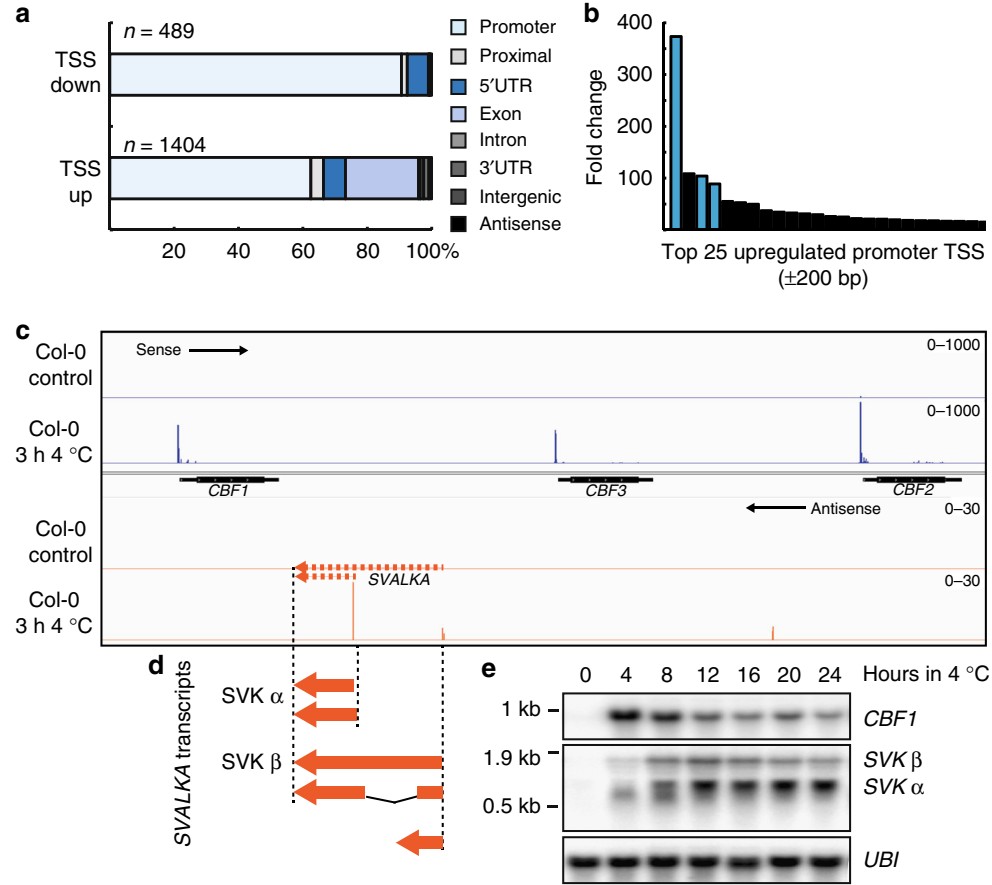

**Fig. 1** Transcription Start Site (TSS) sequencing in Col-0 after 3 h of exposure to 4 °C identified the long non-coding RNA, *SVALKA*. **a** Differentially expressed TSS after cold exposure in Col-0. A total of 1893 TSS changed their expression significantly ($p < 0.05$). Of these, 489 were down-regulated while 1404 were up-regulated. The TSS were classified according to their position relative to gene bodies . The majority of differentially expressed TSS were found in or around promoters. **b** Activity of up-regulated promoters after cold exposure identities the *CBF* genes (indicated in blue bars) as highly up-regulated. Graph represents the fold change of the top 25 up-regulated promoters in Col-0 after cold exposure. **c** Screenshot of the *CBF* genomic region and the identified TSS. The upper panel shows the sense TSS-seq reads direction (blue) and clear TSS can be found for the three *CBF* genes. The lower panel shows the TSS-seq reads in antisense direction (orange) where a group of cold-induced TSS was identified in the intergenic region between *CBF1* and *CBF3*. **d** Summary of 3′- and 5′-RACE of *SVALKA*. Two clusters of capped TSS were found for *SVALKA*, a distal TSS centered around 2360 bp and a proximal TSS centered around 1386 bp in respect to the translation start of *CBF1*. A cluster of polyadenylation sites were found 846–897 bp in respect to the translation start of *CBF1*. The plethora of *SVALKA* transcripts includes a spliced isoform with a splice site from 1386 to 2113 bp in respect to the translation start of *CBF1*. **e** RNA levels of *CBF1* and *SVALKA* during a time course of cold exposure of Col-0. While *CBF1* transiently is upregulated early in the cold response, *SVALKA* responds later and its expression is anti-correlated to sense *CBF1*. *UBI* is used as a loading control. Experiments were done with three biological replicates showing similar results. Uncropped blots can be found in the Source Data file

expression by RT-qPCR. *svk-1* showed reduced expression of *SVK* while the *uns-1* mutants showed slightly elevated levels of *SVK* compared to wild-type. As expected, the *SVK OE* mutant showed greatly increased level of *SVK* (Fig. 2d). Interestingly, we detected *CBF1* mis-regulation in all three mutants (Fig. 2e). The combination of high level of *SVK* expression and increased *CBF1* expression in *uns-1* compared to wild-type argues against a trans-acting function for the *SVK* lncRNA (Fig. 2d, e). In this mutant, *SVK* is expressed 4–5 kb away (compared to 110 bp for wild-type) from *CBF1* but *CBF1* expression is still increased compared to wild-type. Both the *uns-1* and *svk-1* mutants showed an increase of *CBF1* expression in response to cold exposure, while the *SVK OE* mutant lead to decreased *CBF1* levels, supporting the notion that *SVK* plays a repressive role on *CBF1* mRNA levels. We could not see any effect on the neighboring *CBF3* and *CBF2* genes (Supplementary Figure 6). Increased *CBF1* levels in the *uns-1* and the *svk-1* mutants lead to greater induction of *CBF*-activated *COR* genes (Fig. 3a). These findings suggested that *SVK* transcription could have a biologically important role

and affect cold acclimation and freezing tolerance. We conducted an electrolyte leakage test of wild-type, *svk-1* and *uns-1* plants to test the role of *SVK* in freezing tolerance (Fig. 3b). In this test, leaf discs of non-acclimated and cold-acclimated plants are in contact with water and exposed to decreasing freezing temperatures and measured for leakage of electrolytes (i.e., plasma membrane disruption). Surviving cells keep their electrolytes within the cell. Therefore, measuring electrolyte leakage is a measurement of how well the cells can survive freezing temperatures[28]. Cold-acclimated *svk-1* and *uns-1* showed a higher freezing tolerance compared to wild type. Wild-type plants showed an electrolyte leakage of 50% at −6.5 °C (±0.2 °C, 95% CI) compared to *svk-1* (−7.7 °C, ±0.2 °C) and *uns-1* (−7.9 °C, ±0.3 °C). The increase in freezing tolerance is remarkable and is similar to that of the *myb15* mutant, a transcription factor that directly suppresses *CBF* expression[29]. Together, our findings suggest that the genomic region encoding *SVK* represses cold-induced *CBF1* expression and has a biologically relevant effect on cold acclimation and freezing tolerance in *Arabidopsis*.

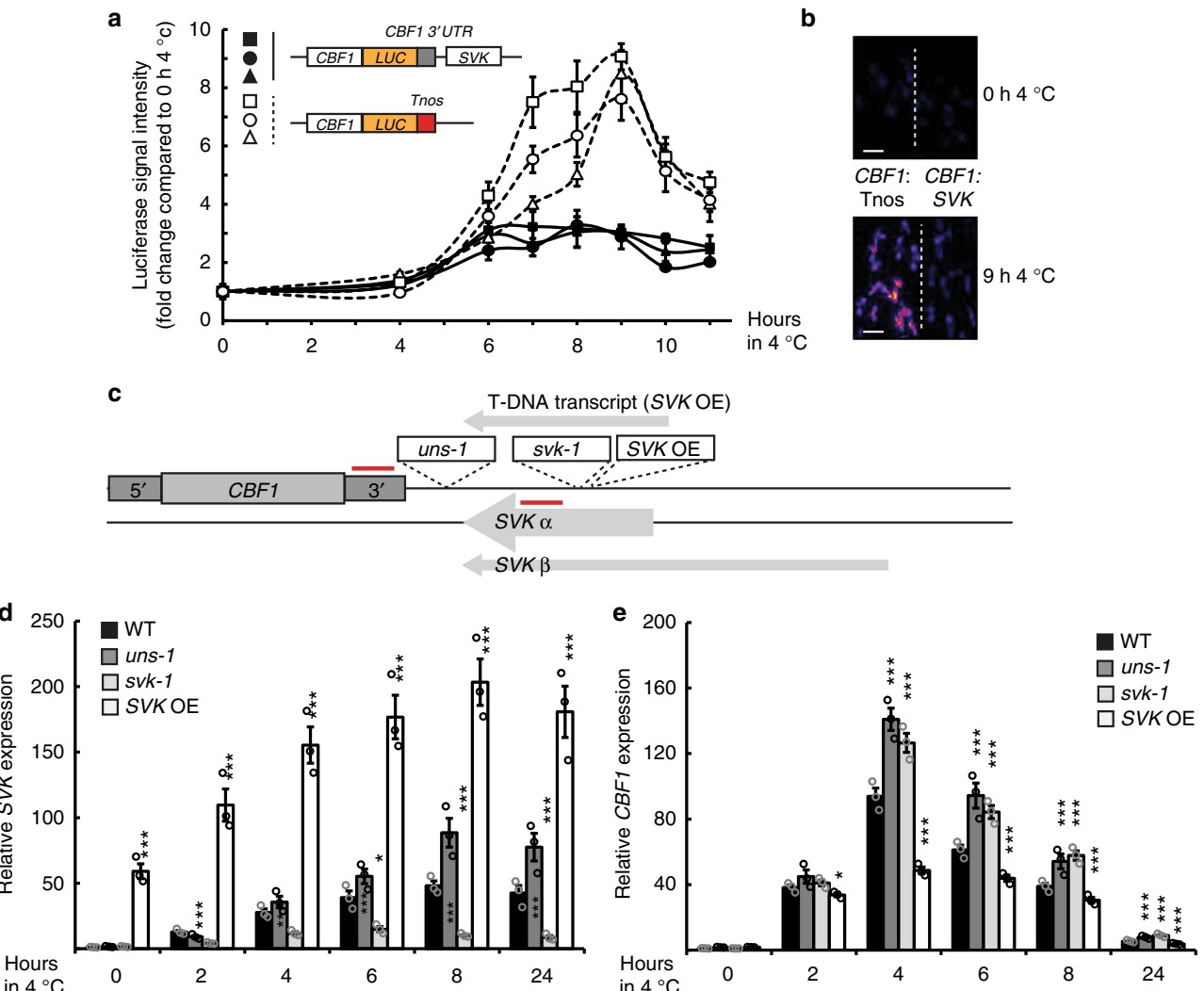

**Fig. 2** Characterization of the long non-coding RNA, *SVALKA*. **a** LUCIFERASE (LUC) activity of two different *CBF1-LUC* constructs. White markers indicate three independent transformants of a construct where *LUC* expression is driven by the *CBF1* promoter and terminated by the $T_{NOS}$ terminator (CBF1:$T_{NOS}$). Black markers indicate three independent transformants of constructs where *LUC* expression is driven by the CBF1 promoter with the endogenous *CBF1* terminator and full *SVALKA* sequence (CBF1:SVK). The CBF1:SVK lines showed a decreased LUC activity following cold exposure compared to the CBF1:$T_{NOS}$) lines. LUC activity was measured as average pixel intensity of at least 5 seedlings for each time point. Markers represent mean with standard deviation. Source data are provided as a Source Data file. **b** Representative images of lines in control conditions and after 9 h of cold exposure for the two LUC constructs. Scale bar represents 1 cm. **c** Graphical representation of the insertion positions of the T-DNA lines used in this study. The qPCR probes are indicated with red and used in (**d**) and (**e**). See Supplementary Figure 5 for additional information on the insertion mutants. **d**, **e** Relative *SVK* (**d**) and *CBF1* (**e**) expression determined by RT-qPCR in WT and mutants that effect *SVK* following exposure to cold. Bars represent mean (black: WT, dark grey: *uns-1*, light grey: *svk-1*, white: *SVK OE*, ±SEM) from three biological replicates (rings). The relative level of *SVK* and *CBF1* transcripts were normalized to the level in WT in control conditions. Statistically significant differences between means were determined with Student's *t*-test (*$p < 0.05$, ***$p < 0.001$). Source data are provided as a Source Data file

**Read-through transcription of *SVK* results in *asCBF1*.** To better understand the mechanism of *SVK* repression, we revisited the effects of the *uns-1* insertion. *uns-1* is inserted downstream of *SVK* yet results in equivalent cold-related defects compared to *svk-1*. A possible explanation for the effects seen in the *svk-1* and *uns-1* mutants would be that read-through transcription of *CBF1* is reduced by the T-DNA insertions that may increase the stability of the *CBF1* mRNA. The *Arabidopsis* 5′−3′ exonuclease XRN3 is associated with the removal of uncapped nuclear transcripts that are expected for read-through transcripts derived from RNAPII termination[30]. Consistently, we find increased *CBF1* read-through transcription in the *xrn3–3* mutant (Supplementary Figure 7). However, we could not detect decreased *CBF1* read-through in the *svk-1* and *uns-1* mutants, arguing against

increased mRNA stability linked to increased transcriptional termination efficiency in these mutants (Supplementary Figure 7). Another possibility that could reconcile our findings would be if *SVK* was promoting transcription of a cryptic antisense transcript into the *CBF1* gene body, since such a transcript would be disrupted in *uns-1* and *svk-1*. To test this hypothesis, we carefully examined the presence of an antisense transcript mapping to the 3′-UTR of *CBF1* (Fig. 4a, b). *HEN2* is part of the nucleoplasmic 3′ to 5′ exosome responsible for degrading many types of non-coding RNAPII transcripts[31]. We used a high resolution time series of cold exposed samples of wild type and *hen2-2* to enable the detection of cryptic antisense transcripts. A *CBF1* antisense transcript (*asCBF1*) corresponding to roughly 250 nt was detectable in the *hen2-2* mutant, yet not in the wild type (Fig. 4a,

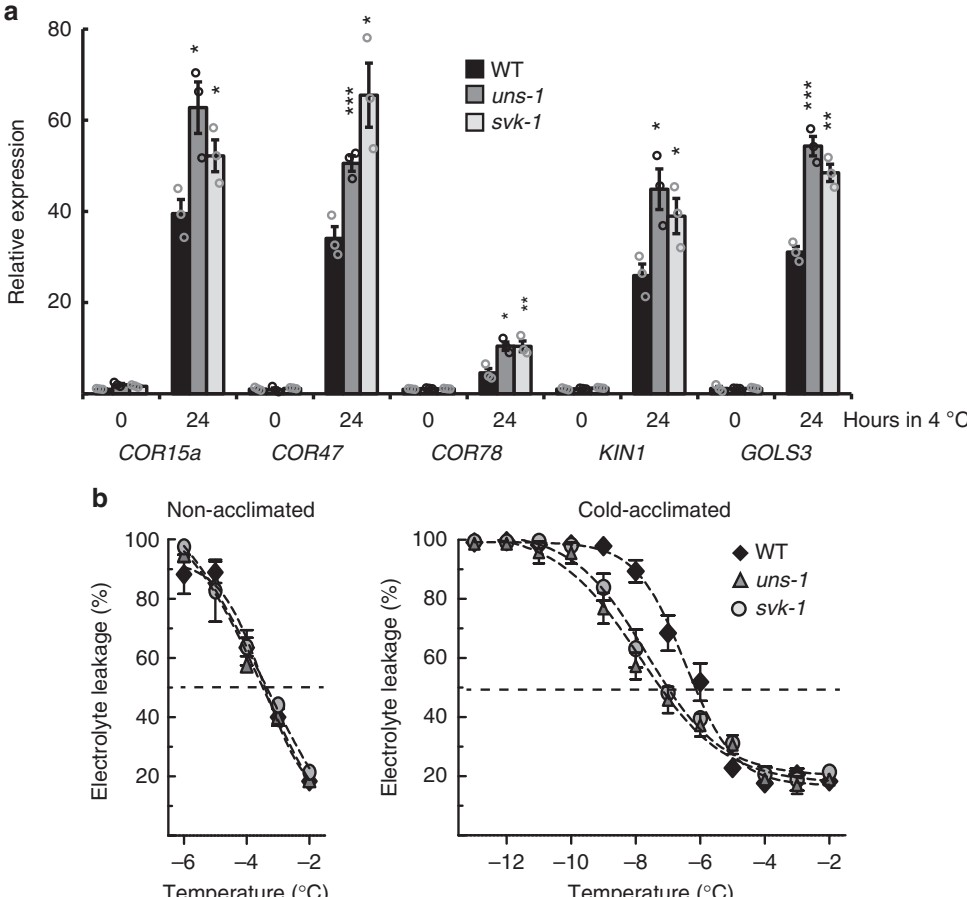

**Fig. 3** Mutants that affect *SVK* show increased freezing tolerance. **a** Relative level of different cold-responsive transcripts in control and after 24 h of cold exposure in WT, *svk-1* and *uns-1* determined by RT-qPCR. Bars represent mean (black: WT, dark grey: *uns-1*, light grey: *svk-1*, ±SEM) from three biological replicates (rings). The relative level of the *COR* transcripts were normalized to the level in WT in control conditions. Statistically significant differences were determined with Student's *t*-test (*$p < 0.05$, **$p < 0.01$, ***$p < 0.001$). Source data are provided as a Source Data file. **b** Electrolyte leakage in non-acclimated (left panel) and cold-acclimated (right panel) WT (black), *svk-1* (light grey) and *uns-1* (dark grey) plants. Leaf discs of each genotype were in contact with deionized water in a test tube and exposed to −2 °C for 1 h followed by a decreased temperature at a rate of 2 °C/h. Samples were taken out at the indicated temperatures and the solution in the tubes was measured for electrolytes. Subsequently, each tube was submersed in liquid nitrogen allowed to thaw and measured again for electrolytes. Each marker represents the mean electrolyte leakage (electrolyte content before exposure to liquid nitrogen/total electrolyte content) after freezing test compared to the total electrolyte content (±standard deviation) from at least three biological replicates. Source data are provided as a Source Data file

Supplementary Figure 8a). *asCBF1* was also detected in additional mutants disrupting the nuclear exosome (Supplementary Figure 8b). However, we could not detect *asCBF1* in the *sop1-5* (*suppressor of pas2*) mutant that have an increase of a subset of HEN2-dependent degradation targets[32] or the *trl-1* mutant, which is disrupted for TRL1 (TRF4/5-like), a factor involved in 3′-processing of rRNA in the nucleolus[33] (Supplementary Figure 8b). Together, these results suggest *asCBF1* transcript is a nucleoplasmic exosome target with expression levels correlated to *SVK*. The size of *asCBF1* suggests it is either a *SVK* read-through transcript that is cleaved at the poly-(A) signal or a new transcription initiation event. To confirm that *asCBF1* depends on *SVK* transcription, we crossed the *svk-1* and *uns-1* mutants to *hen2-2* (Fig. 4c). In both double mutants, the *asCBF1* RNA was lost, confirming that *asCBF1* transcription depends on *SVK*.

Transcription in higher eukaryotes occurs beyond the poly-(A) termination signal before nascent transcript cleavage is triggered[34]. Nascent transcript cleavage creates the 3′-end of the full-length mRNA substrate for poly-adenylation, and a free 5′-end is attacked by 5′ to 3′ exonucleases eliciting transcription termination according to the torpedo model[35,36]. To detect any read-through transcription from *SVK*, we purified RNA associating with RNAPII complexes (Supplementary Figure 9a). To isolate nascent RNA, we used a stable *Arabidopsis* line with a FLAG-tagged version of NRPB2 (i.e., the second largest subunit of RNAPII). After IP, NRPB2 co-eluted with NRPB1, confirming that we purified intact RNAPII complexes (Supplementary Figure 9b-d). Subsequently, we isolated the RNA attached to the purified RNAPII complexes. We profiled this RNA for the presence of *SVK* transcripts extending into *CBF1* as antisense transcripts with qPCR (Fig. 4d). We detected quantifiable levels of a read-through product after 4 h of cold exposure and increased expression after 8 h in the nascent RNA fraction. Consistent with *asCBF1* being a product of read-through following *SVK* 3′-end formation and cleavage, we could not detect the *SVK-asCBF1* read-through transcript in total RNA (i.e. steady state) or in a Mock-IP control samples (i.e. IP with a non-FLAG tagged line). Detection in nuclear exosome mutants and nascent transcript preparation gives *asCBF1* characteristics of a cryptic lncRNA that is quickly cleaved and degraded in wild-type. XRN3 is thought to represent the 5′-to-3′ torpedo exonuclease mediating transcriptional termination in *Arabidopsis*[30,36]. However, we did not detect

the *asCBF1* RNA or *CBF1* mRNA mis-regulation in the *xrn3-3* mutant (Supplementary Figure 10). These data suggest that transcriptional termination of *SVK* read-through is either XRN3 independent, or the hypomorphic *xrn3-3* allele does not disrupt XRN3 function strong enough to detect *SVK* termination defects. In summary, our findings revealed *asCBF1* as a *HEN2*-dependent

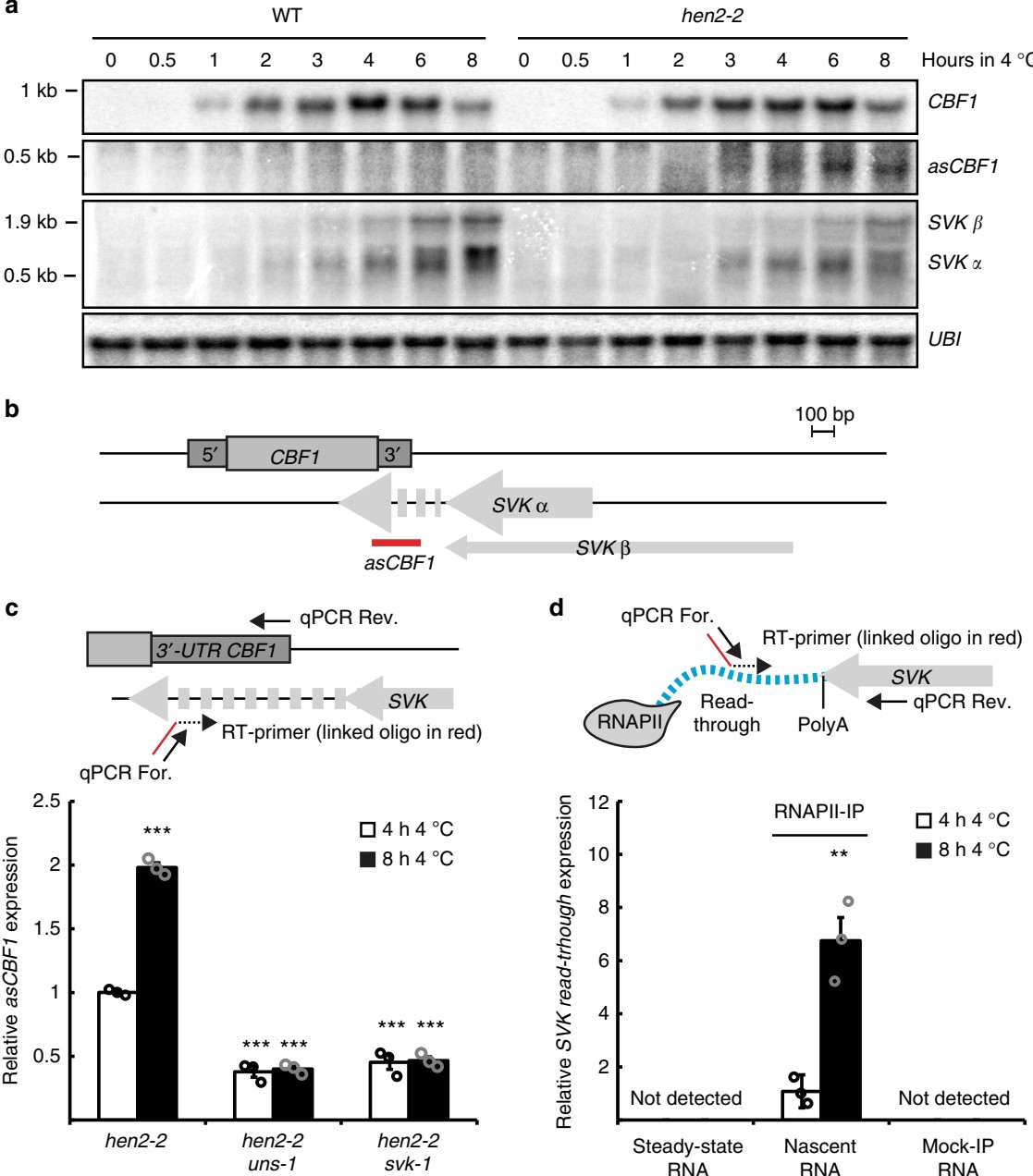

**Fig. 4** *SVALKA* transcription mediates transcription activity antisense to *CBF1*. **a** Representative Northern blot of a cold exposure time series of Col-0 (WT) and *hen2-2*. The probe used for *asCBF1* is shown in **b**. Blots were repeated with three biological replicates with similar results. *UBI* is used as loading control. Uncropped blots can be found in the Source Data file. **b** Graphical representation of the *CBF1-SVK* genomic region. The probe for *asCBF1* is shown in red. RT was done with an oligo-linked primer to ensure strand-specificity and generated cDNA was used in **c**. **c** RT-qPCR of *asCBF1* after cold exposure in *hen2-2* and the double mutants, *hen2-2uns-1,* and *hen2-2svk-1*. The RT-primer and the qPCR primers are shown in the graphical representation above the graph. Bars represent mean (white: 4 h 4 °C, black: 8 h 4 °C, ±SEM) from three biological replicates (rings). The relative level of *asCBF1* was normalized to the level in *hen2-2* after 4 h of 4 °C. Statistically significant differences were determined with Student's *t*-test (\*\*\**p* < 0.001). Source data are provided as a Source Data file. **d**) RT-qPCR of *SVK* read-through transcripts. Hundred nanograms of RNA from an RNAPII-IP (i.e. RNA attached to RNAPII) or total RNA was converted to cDNA with an oligo-linked primer. The RT-primer and qPCR primers used can be seen in the graphical representation above the graph. The RT-primer annealed to a sequence downstream of the PolyA signal of *SVK*. A read-through transcript can be detected in the nascent RNA samples with an increased expression after 8 h of cold exposure. No signal could be seen in the Mock-IP or total RNA sample. Each bar represents the mean (white: 4 h 4 °C, black: 8 h 4 °C, ±SEM) from three biological replicates (rings). The relative level of read-through transcripts was normalized to the level in the nascent sample after 4 h of 4 °C. Statistically significant differences were determined with Student's *t*-test (\*\*\**p* < 0.001). Source data are provided as a Source Data file

"cryptic" antisense transcript that depends on *SVK* read-through transcription during cold to repress *CBF1* induction.

**CBF1 expression is limited by RNAPII collision**. Two molecular mechanisms could explain the observed *SVK*-mediated effects on *CBF1* expression: (1) transcription of *asCBF1* activates the siRNA pathway to trigger repression[37], or (2) the act of *asCBF1* transcription itself causes sense/antisense competition and RNAPII collision[38]. To test if siRNA were involved, we performed cold exposure of several different mutants in various siRNA pathways and probed for *CBF1* mRNA levels (Supplementary Figure 11a). All the mutants failed to show increased *CBF1* levels (Supplementary Figure 11b), ruling out the siRNA model. A RNAPII collision model predicts a discrepancy of transcription between the 5′-end and the 3′-end of *CBF1* caused by stalled RNAPII transcription complexes in the 3′-end of *CBF1*[38]. Our protocol to isolate nascent RNA enabled us to detect strand-specific sites of RNAPII accumulation (i.e., potential pause sites) with RT-qPCR (Fig. 5b, Supplementary Figure 9). The 5′-end probe (Probe 1 in Fig. 5c) showed no significant difference, suggesting that RNAPII occupancy early in *CBF1* elongation was indistinguishable between 4 and 8 h of 4 °C. The probe in the 3′-end of the *CBF1* exon (Probe 2 in Fig. 5c) showed that RNAPII occupancy was significantly increased over this region of *CBF1* after 8 h compared to 4 h. Intriguingly, the increased RNAPII occupancy at the 3′-end of the *CBF1* exon was not detected in the 3′-UTR (Probe 3 in Fig. 5c). These data are consistent with the collision model that predicts RNAPII complexes are pre-maturely terminated before they have time to fully transcribe a full-length *CBF1* transcript.

The 3′-UTR probe assays the steady state level of full-length *CBF1* mRNA (Fig. 2e). The steady state *CBF1* RNA levels after 4 and 8 h of cold exposure showed lower levels of *CBF1* transcripts containing the middle (probe 2 in Fig. 5c) and the end (probe 3 in Fig. 5c) than the beginning of *CBF1* (probe 1 in Fig. 5c). These data confirm that the major decrease after 8 h of cold exposure in full-length *CBF1* mRNA levels relate to differences in the 3′-end. In agreement with these results, we could detect high amounts of pre-maturely terminated *CBF1* when sense transcripts are probed in the 5′-end compared to the 3′-end with Northern blot (Supplementary Figure 12). This effect was exaggerated in the *hen2-2* mutant, suggesting that many pre-terminated *CBF1* events are rapidly degraded by the nuclear exosome. Probes for antisense transcripts of *SVK* showed an increasing RNAPII occupancy over the *SVK* transcription unit following longer exposure to cold (Probe 6 in Fig. 5d). We detected *SVK* read-through transcripts in the nascent fraction (Probe 5 in Fig. 5d). No products were detected with the probe further downstream (Probe 4 in Fig. 5d). These data support that RNAPII complexes are terminated before transcription reaches further into the *CBF1* exon. This is also consistent with the size of the *asCBF1* transcript (Fig. 4a). We complemented these results with total RNAPII quantitative Chromatin Immuno-Precipitation (qChIP) analyses following exposure to 4 °C. Reminiscent of the results from analyzing nascent transcripts, we detected a higher RNAPII occupancy over the *CBF1* exon after 8 h compared to 4 h of 4 °C. In contrast, the probe at the promoter of *CBF1* (pCBF1 in Fig. 5a–e) showed a decrease of RNAPII occupancy after 8 h. These results strongly suggest that RNAPII complexes are stalled in the 3′-end of *CBF1* since full-length *CBF1* mRNA levels decrease after 8 h compared to 4 h of cold exposure (Figs 1e, 5c). qChIP probes for *SVK* showed higher use of the *SVK* promoter following cold exposure that was also seen for the *SVK* transcription unit (Fig. 5f). Another prediction of the collision model is that increased RNAPII occupancy over the *CBF1* exon in wild-type should be abolished in the *uns-1* mutant, since no collision would occur due to the absence of *asCBF1* transcription. Our hypothesis was tested and confirmed by qChIP for total RNAPII over the *CBF1* exon in wild type and *uns-1* since no increase of RNAPII occupancy was observed in the *uns-1* mutant (Fig. 5g).

Collectively, these results imply the presence of stalled RNAPII complexes over the 3′-end of *CBF1* that were correlated with increased *asCBF1* expression. In conclusion, our results showed that read-through of a lncRNA triggers head-to-head RNAPII collision over the *CBF1* gene body leading to regulated termination of *CBF1* transcription and a decrease of full-length *CBF1* mRNA levels (Fig. 5h). This study illustrates how lncRNA read-through transcription limits peak expression of gene expression to promote an appropriate response to environmental change.

## Discussion

Environmental change alters RNAPII transcription in genomes so that organisms can adequately respond to new conditions[5,39]. These transcriptional responses include high RNAPII activity in non-coding regions, resulting in lncRNA transcription. On the one hand, lncRNA molecules may provide function, for example through riboswitch regulation[40,41]. On the other hand, the process of lncRNA transcription by RNAPII can affect the regulation of gene expression in the vicinity of target genes[42]. A curious result of widespread lncRNA transcription is that genes are often transcribed on both DNA strands, resulting in mRNA from the sense strand, and lncRNA from the antisense strand[43].

Antisense transcription is detected in many genomes, and genome-wide analyses suggest that these phenomena can be either positively or negatively correlated with the corresponding sense transcript[44]. The positive correlations between sense and antisense expression represent a puzzling conundrum, as this relationship poses the question of how two converging RNAPII enzymes transcribing the same DNA can move past each other[38]. Sense and antisense transcription pairs on cell population level have been resolved as binary expression states for the individual transcripts at single-cell level[45]. Perhaps alternative expression states of sense or antisense lncRNA transcription in individual cells offers an explanation for why examples of gene repression by RNAPII collision are relatively rare. Overall, antisense transcription appears to be a common strategy to affect gene repression[44].

Our study uncovers how the interplay between two lncRNAs limits the production of maximal mRNA levels of the *Arabidopsis CBF1* gene by an RNAPII collision mechanism in the 3′-end of *CBF1*. Cold-induced transcription initiation in the *CBF* locus triggers expression of a cascade of antisense transcripts that fine-tunes the level of *CBF1* mRNA, thereby representing a negative feedback required for plants to appropriately acclimate to low temperatures.

The identification of interplay between a stable lncRNA *SVALKA* and a cryptic lncRNA *asCBF1* acting together in gene repression represents an important aspect of our study. The cryptic lncRNA *asCBF1* is detectable after 6 h of cold treatment in nuclear exosome mutant backgrounds. *asCBF1* is likely representative of additional lncRNAs with regulatory roles that are challenging to study owing to their cryptic and environmental specific expression characteristics. Our findings of multiple lncRNAs acting together is in line with recent findings suggesting a coordinated effect of multiple lncRNAs to regulate meiosis in yeasts[46,47]. The combinations of multiple lncRNAs amplify the interfaces with regulatory potential and may increase precision of target regulation. Since *asCBF1* results from RNAPII read-through transcription of *SVK*, our findings highlight the process of transcriptional termination of lncRNAs for regulation.

Selective transcriptional termination of the *Arabidopsis* lncRNA *COOLAIR* determines *FLC* expression states by a chromatin-based mechanism[2]. A similar mechanism is more difficult to elucidate in this case due to the relatively small size of the *CBF1* gene body (i.e. 946 bp). Moreover, the class of budding yeast stable uncharacterized transcripts (SUTs) with environment-

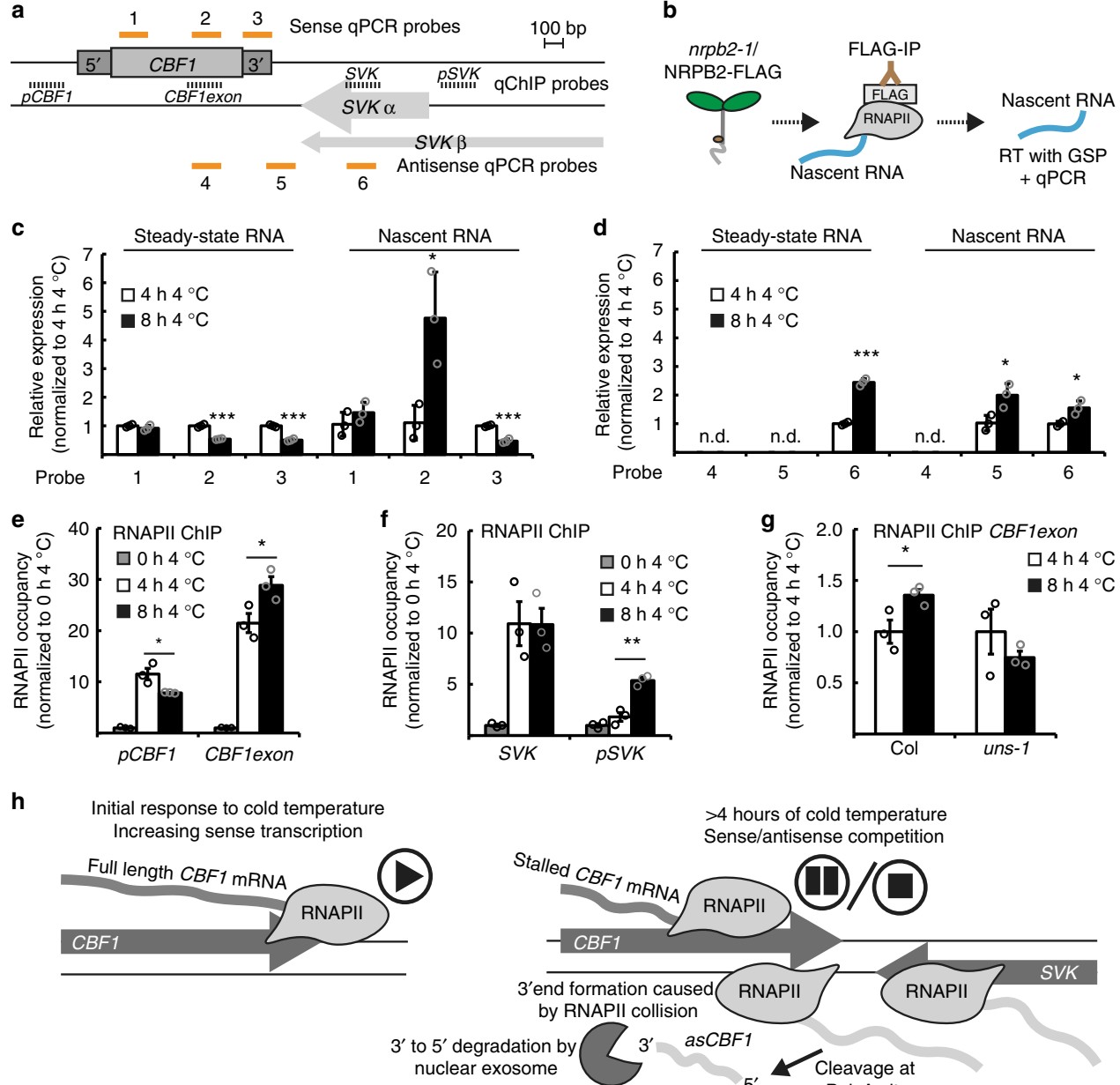

**Fig. 5** Antisense transcription to *CBF1* results in stalling mRNA transcription at the 3′-end of *CBF1*. **a** Graphical representation of the sense and antisense qPCR probes used in **c** and **d**. RT was done with an oligo-linked primer to ensure strand-specificity. qChIP probes used in **e–g** are indicated with a dashed black line. **b** Graphical visualization of the RNAPII-IP to isolate nascent RNA. **c–d**) RT-qPCR with the sense probes (1–3, **c**) and with the antisense probes (4–6, **d**). All probes are shown in **a**. Nascent RNA was compared to a total RNA preparation (steady-state level of RNA). Bars represent mean (white: 4 h 4 °C, black: 8 h 4 °C, ±SEM) from three biological replicates (rings). The relative level of transcripts was normalized to the level at 4 h 4 °C. Statistically significant differences between 4 and 8 h of 4 °C were determined with Student's t-test (*$p < 0.05$, ***$p < 0.001$). Source data are provided as a Source Data file. **e–g** qChIP of total RNAPII along the *CBF1* genomic region following cold exposure in WT (**e**), of the *SVK* genomic region in WT (**f**) and of the *CBF1* transcription unit in WT and the *uns-1* mutant following 4 and 8 h of 4 °C (**g**). The location of promoter (*pCBF1*), *CBF1* exon, the promoter (*pSVK*) and SVK probes can be seen in **a**. Bars represent the mean of three biological replicates (grey: 0 h 4 °C, white: 4 h 4 °C, black: 8 h 4 °C, ±SEM) and are normalized to the level at 0 h 4 °C. Statistically significant differences were determined with Student's *t*-test (*$p < 0.05$, **$p < 0.01$, ***$p < 0.001$). Source data are provided as a Source Data file. **h** Mechanistic model of how transcription of *SVK* represses sense *CBF1* transcription. Initially during cold exposure, *SVK* is not expressed and sense *CBF1* can be transcribed without hindrance (left panel). *CBF1* expression peaks after 4 h of cold exposure. Simultaneously, expression of the lncRNA *SVK* is increased in the antisense direction of *CBF1* (right panel). Read-through transcription of *SVK* results in transcription antisense to the 3′-end of *CBF1* and an increase of RNAPII occupancy on both strands. The intensification of transcribing polymerases in both directions creates RNAPII collision and stalling of *CBF1* sense transcription. The outcome of the collision is that fewer RNAPII complexes reach the end of the *CBF1* sense transcription unit and a decrease of full-length *CBF1* mRNA

specific expression is characterized by inefficient termination resulting in read-through transcription into neighboring genes. Consistently, regulation of yeast gene expression by read-through transcription of SUTs is well-documented[48]. Extended SUT species link transcripts to create interdependency of loci to achieve coordinated expression changes[43]. The effects of the *SVK-asCBF1* circuitry support the idea that cascades of lncRNA transcription provide a mechanism to coordinate local regulation.

lncRNA cascades increase the spacing distance between initial lncRNA transcription events and the target gene, in contrast to a straightforward antisense lncRNA. Multiple cascading transcripts provide more room for independent regulatory inputs to achieve precision in regulation. The topology of the *Arabidopsis* genome is thought to be gene-centric with local gene loops rather than larger topologically associated domains (TADs)[49,50]. We would expect extensive co-regulation if sense and antisense lncRNA transcription were part of the same gene loop. Indeed, alternative gene loop formation is triggered by transcription of the lncRNA *APOLO* to regulate expression of the *PINOID* gene[51]. Read-through transcription of the lncRNA *SVALKA* may provide a mechanism to invade and break positive feedback mechanisms reinforcing *CBF1* transcription. As lncRNA transcription in yeast and mammalian cells correlate with the boundaries of chromatin domains, roles of lncRNA in communicating regulation across domain boundaries may be common despite differences in genomic topologies[52,53].

The entire transcripts in the *CBF* cluster increase expression in response to short-term cold temperatures[11]. The discovery of the *SVK-asCBF1* lncRNA cascade illustrates how transcriptional activation of this region is associated with an inbuilt negative feedback mechanism that limits the timing of maximal *CBF1* expression by triggering RNAPII collision. While *CBF* genes promote freezing tolerance, excess *CBF* gene expression is associated with fitness penalties[16–19]. In this regard, the *SVK-asCBF1* cascade provides a mechanism to tightly control maximal *CBF1* expression and timing that could be exploited to engineer freezing tolerance with mitigated fitness costs. The conservation of *CBF* gene clusters and their role in promoting plant freezing tolerance potentially suggests that lncRNA cascades equivalent to *SVK-asCBF1* could exist in other species. The lncRNA-mediated effects of long-term cold exposure on plant gene expression have been well characterized for the *FLC* gene[2,45]. Our research demonstrates that lncRNAs also mediate the response to short-term and transient cold exposure by timing maximal gene expression. Future research will be needed to investigate if biophysical properties of lncRNA transcription may provide a particularly effective sensor for temperature. Perhaps more likely, lncRNA functions in temperature responses will coincide with the intensity of research efforts addressing this question in plants to prepare for changing climates. In sum, our findings provide a compelling example of functional lncRNA transcription in cells to limit gene induction by triggering RNAPII collisions.

## Methods

**Plant materials and growth conditions**. *A. thaliana* seedlings were grown on ½ MS + 1% Sucrose plates in long day photoperiod (16 h light/8 h dark) at 22 °C/18 °C unless otherwise stated. For all experiments, Col-0 was used as wild-type background. All newly characterized T-DNA lines in this study were identified at [http://signal.salk.edu/cgi-bin/tdnaexpress?gene=] and seeds from the lines were ordered from the Nottingham Arabidopsis Stock Centre. Genotypes used in this study can be found in Supplementary Data 3. For cold treatment, seedlings were grown in control conditions for 10 days (100 µE m$^{-2}$ s$^{-1}$) and subsequently transferred to 4 °C for indicated times and sampled. Light intensity during cold treatment was approximately 25 µE m$^{-2}$ s$^{-1}$. For freezing tests, plants were grown in short day conditions (8 h light/16 h dark) at 22 °C/18 °C for 6 weeks. Cold acclimation was performed in short day conditions at 4 °C for 4 days prior to freezing test.

**Cloning and LUC assay**. Primers for cloning can be found in Supplementary Data 4. For the *CBF1:Tnos* construct, a fragment encompassing the *CBF1* promoter and gene body (−1903 bp to +639 bp relative to the translation start) was amplified from genomic DNA with Phusion polymerase (Thermo Fisher Scientific, USA) and ligated to the pENTR/D-TOPO vector (Invitrogen, USA). After sequencing to eliminate any cloning errors the *pENTR/CBF1* vector was incubated with *pGWB535* vector in a LR reaction. The final vector, *pGWB535/CBF1:Tnos* was transformed into Col-0 plants using the floral dip method . For the *CBF1:SVK* construct, a fragment encompassing the *SVK* promoter and transcription unit (+643 bp to +3410 bp relative to the *CBF1* translation start) was amplified from genomic DNA and fused to a *CBF1:LUC* fragment amplified from *pGWB535/CBF1:Tnos*. The fused fragment was ligated to pENTR/D-TOPO and sequenced. A LR reaction between *pENTR/CBF1:SVK* and *pGWB501* produced the final vector, *PGWB501/CBF1:SVK*, which was transformed into Col-0. Transformed seedlings were selected on Hygromycin plates and their progeny (T2 generation) was used for the LUC assay. For detection of LUC, 10 day old seedlings were treated with cold temperature (4 °C). At indicated times 5 µM D-Luciferin (ZellBio, Germany) was sprayed onto seedlings followed by dark incubation for 30–60 min. LUC was detected using a CCD camera and pixel intensity was determined in ImageJ. The average pixel intensity from at least five seedlings was used to get a mean value and used for statistical analysis (see Source Data file).

**RNA extraction, northern analysis, RACE, and RT-qPCR**. Total RNA was extracted using RNeasy Plant Mini Kit (Qiagen, Germany) according to manufacturer's instructions. Northern analysis was performed as described with minor modifications[48]. Five to 20 µg of total RNA was separated on a 1.2% agarose gel with formaldehyde and 1xMOPS. Gels were blotted overnight onto a nylon membrane and crosslinked with UV radiation. Probes were made in a PCR reaction by incorporating radioactive dTTP ($^{32}$P, PerkinElmer, USA) from a DNA probe template. Membranes were exposed to a phosphorimager screen (GE Healthcare, UK) for 1–10 days depending on the expression level of the transcript of interest. Screens were subsequently scanned with a Typhoon scanner (GE Healthcare, UK). RACE experiments were done with a SMARTer RACE 5′/3′ Kit (Takara, Japan) according to manufacturer's instructions. For RT-qPCR, total RNA was DNase treated with TURBO DNase (Thermo Fisher Scientific, USA) and 1 µg DNase treated RNA was subsequently turned into cDNA as per manufacturer's instructions with Superscript III (Invitrogen, USA) using random primers, oligo dT or gene specific primers depending on the experiment. Diluted cDNA (1:10) was used in a PCR reaction with GoTaq qPCR Master mix (Promega, USA) and run on a CFX384 Touch instrument (Bio-Rad, USA). Data were processed in CFX manager and exported to Excel (Microsoft, USA) for further analysis. Relative expression was calculated and normalized to at least two internal reference genes. All primers used in this study can be found in Supplementary Data 4. All uncropped membranes and all source data for graphs can be found in the Source Data file.

**Electrolyte leakage test**. Electrolyte leakage test was performed[54] with leaf discs of non-acclimated or cold-acclimated plants in glass tubes with 200 µl of deionized water. The tubes were then transferred to a programmable bath at −2 °C (FP51, Julabo, Germany). After 1 h, ice formation was induced and the temperature was slowly decreased (−2 °C/h). Samples were taken out of the bath at designated temperatures and cooled on ice for an hour followed by 4 °C. When all samples were collected, 1.3 ml of deionized water was added and the tubes were shaken overnight at 4 °C. Electrolyte leakage was measured using a conductivity cell (CDM210, Radiometer, Denmark). To get total ion content, tubes were immersed in liquid nitrogen, thawed, shaken again overnight and measured for conductivity. Electrolyte leakage was determined by comparing the measured conductivity before and after the liquid nitrogen treatment. Data was fitted to a sigmoidal dose-response with GraphPad Prism and can be found in the Source Data file.

**TSS-seq and bioinformatic analysis**. The genome-wide distribution of TSSs in wild-type was recently mapped in *Arabidopsis* using 5′-CAP-sequencing[25]. Here, we extend these analyses to 2-week old cold-treated seedlings (3 h at 4 °C). Five micrograms of DNase-treated total RNA were treated with CIP (NEB) to remove non-capped RNA species. 5′-caps were removed using Cap-Clip (CellScript) to permit ligation of single-stranded rP5_RND adapter to 5′-ends with T4 RNA ligase 1 (NEB). Poly(A)-enriched RNAs were captured with oligo(dT) Dynabeads (Thermo Fisher Scientific) according to manufacturer's instructions and fragmented in fragmentation buffer (50 mM Tris acetate pH 8.1, 100 mM KOAc, 30 mM MgOA) for 5 mins at 80 °C. First-strand cDNA was generated using SuperScript III (Invitrogen) and random primers following manufacturer's instructions. Second-strand cDNA was generated with the BioNotI-P5-PET oligo and using Phusion High-Fidelity Polymerase (NEB) as per manufacturer's instructions. Biotinylated PCR products were captured by streptavidin-coupled Dynabeads (Thermo Fisher Scientific), end repaired with End Repair Enzyme mix (NEB), A-tailed with Klenow fragment exo- (NEB), and ligated to barcoded Illumina compatible adapter using T4 DNA ligase (NEB). Libraries were amplified by PCR, size selected using AMPure XP beads (Beckman Coulter), pooled following quantification by Bioanalyzer, and sequenced in single end mode on the following flowcell: NextSeq® 500/550 High Output Kit v2 (75 cycles) (Illumina). For the bioinformatics, all supplementary code for the data analysis pipelines described

below is available at [https://github.com/Maxim-Ivanov/Kindgren_et_al_2018]. The NGS data manipulations were detailed in the 01-Processing_5Cap-Seq_data.sh pipeline. In brief, the custom adapter sequences (ATCTCGTATGCCG) were trimmed from 3′ ends of the 5Cap-Seq (TSS-Seq) reads using Trim Galore v0.4.3. Then 8 nt random barcodes (unique molecular identifiers, or UMIs) were trimmed from 5′ ends of reads and appended to read names using a custom script (UMI_to_Fastq_header. py). The resultant Fastq files were aligned to TAIR10 genome using STAR v2.5.2b in the end-to-end mode[55]. SAM files were sorted and converted to BAM using Samtools v1.7[56]. Reads overlapping the rRNA, tRNA, snRNA and snoRNA genes (obtained from the Araport11 annotation) were filtered out using Bedtools v2.25.0[57]. In addition, multimapper reads with MAPQ score below 10 were removed by Samtools. Morever, we filtered out PCR duplicates by analyzing groups of reads sharing the same 5′ genomic position and removing reads with redundant UMIs (this was done using a custom script Deduplicate_BAM_files_on_UMI.py). Finally, stranded Bedgraph files were generated using Bedtools. For visualization in genomic browsers, the forward and reverse Bedgraph files corresponding to the same sample were combined together and normalized to 1 million tags. For the downstream analysis in R environment, the stranded Bedgraph files were filtered for coverage ≥ 2x and ensured to contain only genomic intervals with 1 bp width (Expand_bedGraph_to_single_base_resolution.py). The detection of 5′ tag clusters (TCs) was described in detail in the 02-Calling_5Cap-Seq_TSS.R pipeline. It makes an extensive use of the CAGEfightR package [https://bioconductor.org/packages/release/bioc/html/CAGEfightR.html][58]. Adjacent TSS separated by not more than 20 bp were merged into TSS clusters. The TSS clusters were annotated by intersection with various genomic features which were extracted from the TxDb.Athaliana.BioMart.plantsmart28 package. We used following definitions: proximal [TSS-500bp, TSS-100bp] and promoters [TSS-100bp, TSS + 100bp]. In addition, TSS were further annotated by genomic location (according to TAIR10 and Araport11): promoter, proximal, 5′UTR, exon, intron, 3′UTR, antisense, or intergenic. In case of conflicting annotations, a single annotation was chosen according to the following hierarchy: intergenic < antisense < intron < exon < 5′UTR < proximal < promoter. Differentially expressed TCs were called using the DESeq2 package.

**Polymerase II IP and nascent RNA purification.** Five grams of seedlings from a line where a NRPB2-FLAG construct covers a lethal *nrpb2-1* mutation (described in ref. [59]) were grinded to a fine powder. NRPB2 encodes the second largest subunit of the RNAPII complex and since the FLAG-tagged protein covers a lethal mutation it ensures that all RNAPII complexes in the line contain a tagged NRPB2. RNA that was attached to RNAPII (i.e. actively transcribed RNA) was isolated via a FLAG-IP. Control Western blots that showed the validity of the approach are shown in Supplementary Figure 9. The Mock-IP sample represents an RNA sample obtained via an IP reaction from a line without a FLAG-tagged NRPB2. For Mock-IP 5 g of Col-0 seedlings were used. Fifteen milliliters of ice-cold extraction buffer (20 mM Tris-HCl pH 7.5, 300 mM NaCl, 5 mM MgCl₂ (+5 µl/ml 20% Tween, 1 µl/ml RNAseOUT, 5 µl/ml 1 M DTT, prot. inhibitor tablet (Roche, 1 tablet for 50 ml buffer)) was added to the powder and 660 U of DNase I was subsequently added to the mix. After 20 min on ice, the mixture was put into a centrifuge for 10 min (4 °C, 10,000 g) and the supernatant was recovered. The supernatant was filtered through a 0.45 µm filter and added to Anti-FLAG M2 magnetic beads (Sigma-Aldrich, USA). Supernatant and beads were then incubated on a slowly rotating wheel at 4 °C for 3 h. Beads were washed eight times with extraction buffer and immunoprecipitated polymerase complexes were eluted with 2 mg/ml 3xFLAG peptide (ApexBio, USA) for two times 30 min. Purified polymerase complexes was disrupted and attached RNA was extracted by the miRNeasy Mini Kit (Qiagen, Germany) according to manufacturer's instructions. Hundred nanogram isolated RNA was used to create cDNA with Superscript III (Invitrogen, USA) and gene specific primers.

**Western blotting.** Protein samples were separated on a 4–15% TGX gel (Bio-Rad, USA) and blotted onto a PVDF membrane. Membranes were blocked in 5% milk powder dissolved in PBS. Primary antibody was incubated with membrane overnight at 4 °C. Following morning, membranes were washed in PBS + T two times for 10-min and incubated with secondary antibody (Agilent, USA) for 60 min at room temperature. After three times of 5-min washes in PBS+T proteins were detected with Super-Signal West Pico Chemiluminescent (Thermo Fisher Scientific, USA) and developed with a ChemiDoc MP instrument (Bio-Rad, USA). Following antibodies were used in this study: Monoclonal Anti-FLAG M2 antibody (F3165, Anti-Sigma-Aldrich, USA) and Anti-RBP1 antibody (ab140509, abcam, UK). Uncropped gel pictures can be found in the Source Data file.

**Quantitative chromatin immunoprecipitation (qChIP).** *Arabidopsis* seedlings were grown for 12 days on 1/2 Murashige and Skoog (MS) plates containing 1% sucrose and supplemented with 0.5% Microagar in climate chambers at 22 °C with a 16 h light/8 h dark cycle. For cold-treatment, seedlings were grown in control conditions for 12 days and subsequently transferred to 4 °C for indicated times. Cold-treated seedlings were quickly harvested at 4 °C. Fixation of control and cold-treated seedlings were performed at room-temperature using 1% formaldehyde/PBS and vacuum infiltration. Fixation was stopped by the addition of glycine. qChIP experiments were performed essentially as described in ref. [60], with minor modifications. For immunoprecipitations, Protein A magnetic beads (GenScript) and 2 µg of an antibody (Anti-Total RNA polymerase II 8WG16, ab817, abcam,

UK) were added to solubilized chromatin. Quantitative analysis was performed on captured DNA by qPCR (Bio-Rad). See Supplementary Data 4 for oligonucleotide sequences. ChIP enrichments were calculated as the ratio of product of interest from IP sample to the corresponding input sample. Error bars represent standard error of the mean resulting from at least three independent replicates. Source data can be found in the Source Data file.

**Code availability.** All supplementary code for the data analysis pipelines described is available at [https://github.com/Maxim-Ivanov/Kindgren_et_al_2018].

## Data availability
TSS-seq data is available at NCBI GEO database with accession code GSE119304. TSS-seq analysis used computational methods. The scripts are available at github: [https://github.com/Maxim-Ivanov/Kindgren_et_al_2018]. The authors declare that all other data supporting the findings of this study are available from the corresponding author upon request. The source data underlying all figures can be found in the Source Data file. A reporting summary for this Article is available as a Supplementary Information File.

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

## Acknowledgements

We would like to thank Prof. Åsa Strand and Sofie Grönlund for assistance with the freezing test. We thank Prof. Peter Brodersen for sharing seed and assistance with the Luciferase assay. We thank Prof. Vicent Pelechano for help with the 5′CAP-seq. We thank Jasmin Dilgen for technical assistance, Jan Høstrup for plant care and members of the S.M. laboratory for critical reading of the manuscript. Research in the laboratory of S. M. is supported by a Hallas-Møller Investigator award by the Novo Nordisk Foundation NNF15OC0014202, a Copenhagen Plant Science Centre Young Investigator Starting grant. In addition, this project has received funding from the European Research Council (ERC) and the Marie Curie Actions under the European Union's Horizon 2020 research and innovation programme StG2017-757411 (S.M.) and MSCA-IF 703085 (P.K.). R.A. is supported by an EMBO LTF (ALTF 463–2016).

## Author contributions

P.K., R.A., and M.I. performed the experiments. P.K., R.A., and M.I. analyzed the data. S. M. supervised the project. P.K. and S.M. wrote the manuscript.

## Additional information

**Competing interests:** The authors declare no competing interests.

