## [Peer Review File · Nature Communications]

Point-by-point response to reviewers comments for manuscript: “Transcriptional read-through of the long non-coding RNA SVALKA governs plant cold acclimation” NCOMMS-18-11218-T.

We include the comments in *blue* and our response in **black** below each point.

Reviewer #1:

Authors demonstrate that non-coding RNA SVALKA affect the timing of maximal CBF1 expression and freezing tolerance. Read-through transcription of SVALKA results in overlapping CBF1 on the antisense strand, asCBF1. Thus, CBF1 is suppressed by RNA polymerase II collision with asCBF1. To confirm this hypothesis, several points should be modified.

1. Authors isolated two T-DNA insertion mutants, uns-1 and svk-1. In these mutants, how about expression of CBF3 or CBF2? Because uns-1 is from Salk collection. In the T-DNA of Salk collection, promoters are inserted. Thus, sometimes, overexpression line was obtained from Salk collection. If CBF3 are overexpressed, it is logical that uns-1 exhibits cold tolerance.

T-DNA insertion mutants are a plant-specific functional genomics resource that we did not adequately explain in the initial manuscript. We enhance our manuscript with revised Fig S5 to add clarity and the detailed information about the mutants we have used in the study. We would like to clarify that the mutants are from different collections: the uns-1 insertion is from the SALK collection, while the svk-1 insertion is from the GABI collection. Fig. S5a annotates the strong, constitutive promoters inserted into the CBF genomic region as part of the T-DNA integration event. For example, the NOS and 35S promoters in SALK lines, and 35S and 1'-2' promoter in GABI lines. However, we could not find any strong transcription events originating from the T-DNA in the uns-1 nor the svk-1 mutant with Northern probes 3'- and 5' of the T-DNA (Fig. S5d).

We used qPCR to test if the insertion of promoters in the used T-DNA insertion events affected the expression of the CBF2 and CBF3 genes. If CBF2 and CBF3 expression was affected by the constitutive T-DNA promoters, CBF2 and CBF3 over-expression is expected to be independent of cold treatment. We found no evidence for constitutive CBF2 or CBF3 expression as would be expected if transcriptional control would be influenced by the strong and constitutive promoters in the T-DNA insertions (Fig. S5a). To test if the promoter insertions may amplify the transcription response to cold, we measured CBF2 and CBF3 expression by qPCR prior to the effects mediated by the SVK-asCBF1 cascade. As SVK-asCBF1 is not yet induced at 2h 4C, this experiment allowed us to distinguish effects by the constitutive T-DNA promoters and the SVK-asCBF1 cascade. We found no statistically significant expression differences of CBF2 and CBF3 between wild type and the svk-1 and uns-1 insertion mutants (Fig S6). Collectively, these results support the validity of using svk-1 and uns-1 to dissect cis-regulation of CBF1.

2. Or these T-DNAs may block read-through of CBF1. I mean SVALKKA is not important, just blocking of read-through of CBF1 may increase cold tolerance.

We agree that T-DNA insertions could block read-through transcription from CBF1. 3'-extension read-through arises when RNAPII continues to transcribe downstream of the termination signal. In Arabidopsis, read-through transcription is counter-acted by XRN3, a nuclear localized 5'-3'exoribonuclease. Arabidopsis XRN3 is a homolog to yeast and human Rat1/Xrn2 proteins.

To address a connection between read-through transcription and CBF1 expression, we measured CBF1 read-through transcription levels in Fig. S7. We used an oligo-linked RT-primer to strand-specifically generate cDNA of 3'-extended CBF1 transcripts (Fig. S7). We compared wild-type Col, svk-1, uns-1 and xrn3-3. The xrn3-3 mutant accumulates read-through transcripts in Arabidopsis and serves as positive control for read-through transcription (Kurihara et al., G3, Vol 2, Issue 4, Pages 487-498, 2012). As expected, we detect increased CBF1 3'-extensions in xrn3-3 mutants. Importantly, this did not affect CBF1 mRNA levels in the xrn3-3 mutant (Fig. S10). The uns-1 and svk-1 mutants both showed an increase of CBF1 read-through compared to wild type. Importantly, the increase of read-through transcription in uns-1 and svk-1 insertion mutants rules out the concern that these mutants may decrease read-through transcription (or increase termination). These results are consistent with a role for the SVK-asCBF1 cascade to increase transcriptional termination of the CBF1 gene during the cold.

3. In Fig. 2a, authors need to use CBF1-LUC-Tnos-SVK as a control. If read-through of SVK is important, asTnos may be produced. If asTnos suppresses CBF1 expression, SVK may be required for suppression of CBF1.

Antisense transcription from endogenous terminators is a common feature in eukaryotic genomes. We chose the T_{NOS} terminator to minimize antisense transcription. A similar approach has been used by the Swiezewski lab to disentangle the role of antisense transcription of the DOG1 gene in Arabidopsis (Fedak et al., PNAS, Vol 113, Issue 48, Pages E7846-E7855, 2016). However, even T_{NOS} sequence may trigger antisense transcription that may affect CBF1 transcription.

To address this concern, we quantified antisense transcription from two different LUCIFERASE constructs with qPCR (Fig S4). We used an oligo-linked RT-primer that annealed to the 3'-end of the LUCIFERASE gene to generate strand-specific cDNA of antisense transcripts. In our qPCR reaction, we used a forward primer that annealed to the linked oligo of the RT-primer together with either a reverse primer that annealed to the T_{NOS} sequence or the endogenous CBF1 3'-UTR. The amplified PCR products measure antisense transcription of the different LUC constructs. For the CBF1:SVK construct, we saw a cold-responsive increase of antisense transcription, similar to what we showed for the endogenous CBF1 locus. However, the CBF1:T_{NOS} construct showed less, albeit detectable antisense transcription. Importantly, antisense transcription from the T_{NOS} terminator is not cold-responsive, strongly suggesting that

the antisense transcripts are not involved in regulating the LUC expression. Together, these results showed the validity of our approach. In light of constitutive low-level antisense transcription from the T_{NOS} terminator, we have not tested the suggested CBF:T_{NOS}:SVK construct. The interpretation of effects that may be detectable in CBF:T_{NOS}:SVK would be confounded by the interaction between several sources of potential antisense transcription. All in all, our additional experiments support our findings that SVK drives the cold-induced antisense transcription to regulate LUC expression during cold, consistent with what we present for the endogenous CBF1 locus.

4. To confirm whether read-through of SVALKKA affects expression of CBF1, authors need to make overexpression lines harboring several kinds of SVALKKA (for example, long version, spliced version, proximal version, or read-through version of SVALKKA). If the hypothesis is true, overexpression of read-through version of SVALKKA may significantly suppress expression of CBF1.

There appears to be some confusion that we would like to clarify. The suggested over-expression of lncRNA isoforms is most informative for trans-acting lncRNAs. Trans-acting lncRNAs are derived from one genomic location but act at a distant location. We would like to point out that SVK expression still occurs in the uns-1 mutant. In this mutant SVK expression is moved 4-5 kb away from CBF1 due to the inserted T-DNA, but CBF1 repression is still increased. These results strongly support a cis-acting mechanism for SVK-asCBF1. We enhanced our manuscript by pointing this out more clearly in the text (row 123-125 in the manuscript file with highlighted changes).

Nevertheless, we agree with the reviewer that an overexpression line of SVK could be valuable to further support our model. Unfortunately, targeted “knock-in” of constructs in a desired genomic location is still technically very challenging in Arabidopsis, making a cis-acting over-expression line is difficult to generate. In an attempt to achieve cis-localized over-expression of SVK, we characterized additional T-DNA mutants in the SVK locus. As Reviewer #1 stated in Point 1, T-DNA lines harbors strong promoters within the inserted DNA that sometimes drives transcription that continues over the borders of the T-DNA. Fortunately for us, one of the mutants we characterized showed robust transcription from the T-DNA that continued over the SVK sequence. We named this mutant SVK over-expression (SVK OE) and enhanced our manuscript with experiments using this line to test effects of constitutive cis-restricted SVK over-expression. Key characterization of this line is presented in Fig. S2 and Fig. S5. SVK expression in this mutant is shown in Fig 2 and Fig S5. In the SVK OE mutant, transcription over the SVK sequence is increased 50-200 fold compared to wild type depending on the time of cold treatment.

The SVK OE mutant allows us to test the proposed experiment: What is the effect of CBF1 expression when SVK is strongly up-regulated? Our findings show that CBF1 expression is reduced compared to Col when SVK is over-expressed in cis. These new results support the

notion that SVK is a negative regulator of CBF1 expression and therefore strengthen our revised manuscript.

5. I am wondering why in the svk-1 mutant, expression of CBF1 was not high, compared to wild type? According to Fig. 2e, expression of CBF1 in svk-1 was high only 6 hour after cold treatment. But, because SVALKKA is expressed 2 hour after cold treatment (according to Fig. 3a) and expression level of SVALKKA was very high (Fig. 2d), SVALKKA may suppress expression of CBF1 2, 4, and 8 hour after cold treatment. But difference of expression, compared to wild type, is only 6 hour after cold treatment (Fig. 2e). I wonder whether SVALKKA really affect CBF1 expression or not.

We acknowledge that our initial manuscript did not show a CBF1 over-expression convincingly in the mutants. While we appreciate Northern blots to detect different isoforms of transcripts, it is not the best method to quantify differences in gene expression. Therefore, we have addressed this comment by exchanging Northern blot quantification for qPCR data in Fig. 2. The differences between Col, uns-1 and svk-1 are much more clearly visible with qPCR.

6. In Fig. 3, the blot of asCBF1 needs to be clear. It is difficult to see the band.

To address this comment, the northern blot data has been removed and replaced with qPCR data that are presented in Figure 4c. The results show that the expression of asCBF1 is significantly lower and not cold-responsive in the uns-1 and svk-1 mutants compared to Col. These new results corroborate our conclusions since qPCR data show the requirement of SVK transcription to generate asCBF1, and allow a better quantification of the effect.

7. In Fig. 2f, to confirm whether CBF1-regulon genes are controlled by SVALKKA, expression of other cold-responsive genes, such as KIN1, RD29A, P5CS2, and so on, should be examined.

To address this comment, we added qPCR data for more downstream genes in the revised Figure 3 (COR78, KIN1 and GOLS3). These new data show the same trend as COR15a and COR47 (i.e. significantly higher expression in the uns-1 and svk-1 mutants compared to Col).

Reviewer #2:

Cold treatment of Arabidopsis thaliana is known to induce transient expression of CBF (C-repeat/dehydration-responsive element binding factor family) genes. However, to date, the mechanism that limits CBF expression is unknown. This study aims to provide a mechanism for CBF1 expression regulation. After performing a range of RNA mapping experiments (starting with a Cap-seq approach)

in cold treated samples, the lncRNA SVALKA (Swedish for cold) is identified, which is expressed convergent, but apparently non-overlapping with CBF1. In subsequent mechanistic studies it is suggested that SVALKA is extended upon cold adaption into an unstable transcript that ultimately interferes with sense transcription of CBF1. Thus, whilst cold treatment initially induces of CBF1 transcription, SVALKA transcriptional read-through subsequently reduces this CBF1 activation.

The data presented suggests a negative feedback loop mechanism that supports fast regulation of CBF gene expression. I consider that this study to clearly be of interest to the plant and wider molecular biology field. However, I feel that some of the transcript mapping and characterization of SVALKA needs to be experimentally extended to firm up the paper's conclusions.

We extended the manuscript with additional experimental data as suggested to firm up our conclusions.

My detailed comments are as follows (based on Figures)

Figure 1:

1c: In the position of SVALKA should be marked at least with a dotted arrow Also the transcription direction of CBF genes should be more clearly indicated. The anti-sense profile is quite noisy with apparently only one peak, the SVALKA proximal promoter product, above background? Are the Y axes here the same for sense and antisense profiles?

We thank Reviewer #2 for this comment and made the proposed changes to clarify the manuscript. We enhance the annotation of Fig. 1c to show the TSS of SVALKA more clearly. We annotated the scale of the sense and antisense Y-axes more clearly in the revised Fig. 1. The scale of SVK initiation is very low compared to sense CBF initiation partly due to the relatively short cold exposure (3 hours) in the TSS-seq experiment. SVK reaches its peak of expression after 12-16 hours and is just starting to be expressed at 3 hours. To further clarify Fig 1, we have renamed the isoforms of SVK. SVK α is the major isoform from the proximal promoter and SVK β is the longer isoform originating from the distal promoter. Please see Fig. S3 for quantification of the SVK isoforms with qPCR.

1d,e: It is claimed that no SVALKA transcript overlapping with the 3'UTR of CBF1 is present (based on 3'RACE - not shown). This data should be shown. I wonder what the larger product above SVK on the northern blot is? Looks very similar to the read-through RNAs described in later figures. Perhaps oligo directed RNase H northern could be used to address this issue.

This comment inspired us to improve the clarity of the presentation of RACE results. We enhanced the manuscript with a new supplementary figure (Fig. S2) where the RACE-products

are shown down to nucleotide resolution. We could not detect any longer transcripts that overlap the 3'UTR of CBF1 in our RACE analysis of SVK. The two main SVK transcript isoforms detected by northern blotting correspond to the α - and β - isoforms that are more clearly annotated in revised Figure 1 (and Fig. S2). The longer transcript seen in our Northern blots of SVK represents the long transcript which we have renamed SVK β (Fig. 1). We have also added a new supplementary figure (Fig. S1) that shows an analysis confirming that we could not detect any stable transcripts downstream of the PolyA site of SVK in Col.

Figure 2:

2c: Some clear indication of exactly what the T-DNA inserts comprise is needed (especially for non-plant experts)

We enhanced our manuscript by adding a new supplementary figure with a more in-depth description of the transfer DNA inserted in the mutants (Fig. S5). A precise graphical overview of the T-DNA insertion position down to the nucleotide is shown in the revised in Fig. S2.

2d: the effects of uns-1 and svk-1 mutations on CBF1 levels appear fairly marginal. Why not do longer time points up to 24h as in Figure 1? Again a longer SVK band is evident. Is this read-through transcription? Elektrolyte leakage needs better explanation? i.e. g: not understandable to non-plant expert.

We thank Reviewer #2 for this suggestion and have added a 24 hour time point in our analysis (Fig. 2). We have exchanged Northern blot results with qPCR data since it is better suited for quantitative analysis of gene expression. The differences between Col and the mutants are detected more clearly in the revised qPCR data.

We address the question about the 2 bands detected by northern blotting by enhancing the clarity of SVK transcript isoform annotation in revised Fig. 1 and Fig. S2.

The electrolyte leakage method has now been better explained in the text (row 132-136 in the manuscript file with highlighted changes), methods and in the figure legend of Fig. 3. Briefly, the electrolyte leakage assay measures the integrity of cell membranes in leaf discs in response to decreasing temperature. Disrupted membranes release the cell's electrolytes and this is measured and compared to the total electrolyte content of the leaf disc. The freezing tolerance of a genotype is given at the temperature when 50% of the total electrolyte content is in solution. This test is commonly used to assess freezing tolerance of plant cells (Thalhammer et al., Methods in Molecular Biology: Plant Cold Acclimation, Volume 1166, 2014).

Figure 3:

I feel that the efforts here to detect and define SVK read-through transcripts are somewhat limited by the technology applied. What is really needed here is a nascent transcript mapping approach across

the whole locus such as GRO-seq, NET-seq or ThioU-seq under WT or cold treatment conditions. Also, the simple northern blot (3a) clearly shows the same longer RNA species appearing over the cold treatment time course with or without Hen2 (Exosome). Is this the read-through transcript? 3c,d: don't look very convincing to me.

We addressed this comment with the addition of nascent transcription data across this locus (Fig. 4d, Fig.5). We compare three different RNA samples: 1) steady-state RNA (i.e. total RNA preparation from wild-type), 2) Nascent RNA (i.e. RNA attached to RNAPII isolated via RNAPII-IP), and 3) Mock-IP RNA. We have updated the method section to clearly explain our approach. In addition, we have added a simple workflow diagram of the isolation in Fig 5b and a more detailed flow-chart in Fig. S9. Our results showed that we could only detect SVK read-through in the nascent RNA fraction suggesting that any 3'-extended transcripts of SVK is quickly degraded in wild-type. Importantly, the read-through transcript is cold-responsive and its expression is correlated with the expression of SVK and asCBF1. These results obtained by analyzing nascent transcription strongly support our model where SVK transcription generates a cryptic read-through RNA antisense to CBF1.

We hope the confusion about the SVK transcripts identified by northern blotting could be addressed by our transcript annotation in revised figures Fig. 1d, Fig. S1 and Fig. S2. There are two SVK isoforms (α and β) that we detect by RACE and northern blotting. The longer, SVK- β isoform does not overlap with CBF1 or asCBF1, which is now indicated more precisely in Fig. S2. SVK (α and β) isoforms are no HEN2 targets. In contrast, we can detect the asCBF1 lncRNA in hen2 mutants and nascent transcript analysis. The level of asCBF1 was relatively low, even in the hen2-2 mutant. The low level challenged reliable quantification by Northern blot. To improve asCBF1 transcript quantification, we used a strand-specific oligo-linked RT primer followed by qPCR. The new results clearly showed a significant cold-induction of the asCBF1 transcript in the hen2-2 mutant (Fig. 4c). In the double mutants, hen2-2uns-1 and hen2-2svk-1, the asCBF1 transcript is not induced by cold and was detected at a significantly lower level compared to the hen2-2 single mutant (Fig. 4c). These results show that the generation of asCBF1 relies on cis-acting SVK transcription. As asCBF1 does not contain SVK sequences, we annotate it as separate lncRNA even though asCBF1 production requires SVK transcription.

Figure 4:

Again, I feel the data quality and methods applied limit the interpretation of these data.

b: Simply probing steady state transcript levels over the 5' and 3' ends of CBF1 under cold conditions is too crude a measure to justify a model of 3' end polymerase collision between CBF1 and SVK. Again, a nascent transcription approach is needed here. Also, possibly RNA Polymerase ChIP analysis with multiple locus probes (or ChIP-seq) would be useful.

We thank Reviewer #2 for pointing this out and the experimental suggestions. To better visualize the nascent method used, we have now compared steady-state RNA levels to nascent RNA levels (i.e. total RNA versus RNA attached to purified RNAPII complexes) in a new Figure 5. We have also added more qPCR probes along the CBF1-SVK region to increase the resolution of our assays. These new data clearly show an increase of RNAPII-attached RNA in the 5'-end of the CBF1 exon after 8 hours of cold treatment compared to 4 hours. The probe located further downstream shows a decrease of nascent RNA after 8 hours, suggesting that many transcriptional events do not reach the 3'-UTR of CBF1 after 8 hours of cold treatment. The steady state level of RNA showed no significant difference in the 5'-end of the CBF1 exon but a significant decrease further downstream at 8 hours compared to 4 hours of cold treatment. There is a clear discrepancy between steady-state RNA and nascent RNA at the 3'-end of the CBF1 exon (probe 2 in Fig. 5c). Nascent RNA associated to RNAPII is thus not fully converted to mRNA. These data suggest that RNAPII traveling over the CBF1 gene body slows down/stalls at this position after 8 hours of cold treatment. The antisense probes showed an increase after 8 hours compared to 4 hours which reflected the increase of SVK/asCBF1 transcription. We could only detect SVK read-through in the nascent RNA fraction, confirming that this transcript is rapidly degraded in wild-type. We could not detect any antisense transcripts further downstream (probe 4 in Fig. 5d), suggesting that most RNAPII complexes are terminated before reaching this sequence.

To further address this comment, we performed RNAPII ChIP (Fig. 5). Our new results echo the conclusions derived from nascent RNA analysis. The ChIP experiments were done with an antibody that recognizes the total RNAPII pool. The relatively small size of the CBF1 gene (~1 kb) limits the experimental design. Nevertheless, we could design qChIP probes for the promoter of CBF1 and the CBF1 transcription unit as well as the proximal promoter and transcription unit of SVK (Fig 5a). Similar to what we saw in the nascent RNA qPCR, we detected an increase of RNAPII occupancy over the CBF1 transcription unit after 8 hours compared to 4 hours of cold treatment. This was in contrast to the CBF1 promoter where RNAPII occupancy decreased after 8 hours. It is also in contrast to the level of CBF1 mRNA at 4 h and 8 h, suggesting that RNAPII complexes are stalled/slowed down over the CBF1 transcription unit after 8 hours of cold treatment.

To further expand on these results, we have included RNAPII ChIP for the *uns-1* mutant after 4 and 8 hours of cold treatment for the CBF1 transcription unit (Fig. 5). The increase of RNAPII occupancy in the CBF1-body that we saw in wild type was absent in the *uns-1* mutant. These data suggest that the localized increase of RNAPII (i.e. potential stalling site) was, at least partly, due to the presence of the SVK-asCBF1 cascade. All in all, the new data strengthens our hypothesis that RNAPII stalls as a result of RNAPII collision over the CBF1 gene body in response to SVK transcription.

Finally, if a polymerase collision mechanism is to be invoked then based on yeast studies (e.g. Hobson

et al Mol Cell 2012 48, 365-Svedstrup lab) polymerase ubiquitination may occur to promote its degradation.

We appreciate this very insightful suggestion. We have tried this experiment. Unfortunately, we could not get the necessary double IP ChIP experiments to work. The RNAPII ChIP signal was relatively low after the first IP, which challenged obtaining reliable results from a subsequent UBI-IP.

Reviewer #3:

CBF1 is one of the key regulators for environmental stress responses of Arabidopsis, and thus its gene regulation is physiologically important. The authors have discovered a novel type of gene regulation through transcription of the antisense strand. As shown in the results, the discovered down regulation contributes reduction of the tailing response after 8 h of the cold treatments. One nice point of the report is detection of cryptic RNA species which is not present in WT. Although evidence for the proposed regulation mechanism, transcriptional suppression by Pol II collision through antisense transcription, is not strong enough for its proof, I expect high impact of this work. I have a very positive impression on this report, and thus encourage resubmission after revision.

We thank reviewer #3 for the positive evaluation of the expected impact. The additional data added as a result of the revision further strengthen the case of an example for gene regulation by a RNAPII collision mechanism.

Major point

Fig. 4:

Different profiles between panels c and d seem to imply detection of distinct transcript species between them, because Pol II-IP experiments do not contain any digestion or fragmentation steps. With the current data (Fig 4 a-d), it is not easy for us to speculate why panel d shows a very different profile from the others. I hope the authors provide additional data giving some hint. One possibility might be partial digestion of the sense transcripts in a stalled Pol II-dependent manner, but this possibility apparently does not show good agreement with results in panel b, so it might not be a good model. Additional data is necessary.

To address this comment, we enhanced the manuscript with data from additional experiments and a clearer explanation of our approach. We increased the resolution of the RNAPII-IP-qPCR (i.e. nascent RNA analysis) and added RNAPII ChIP data to map the position of RNAPII across the locus. In the new Fig 5, we compare the steady-state level of RNA (i.e. total RNA preparation) with nascent RNA. We have updated the method section of the RNAPII-IP to more clearly explain our approach. We isolated nascent RNA to be able to strand-specifically see the dynamics of RNAPII occupancy along the CBF1-SVK region. The strand-specificity and increased resolution (i.e we can design qPCR probes in a much closer interval) of this approach is a clear

advantage in comparison with other methods such as ChIP. To get strand-specific cDNA, we employed RT-primers with a linked oligo. The linked sequence is absent from the Arabidopsis genome, circumventing genomic DNA contamination. An additional 3'UTR probe was included in the revised manuscript to detect RNAPII that is prematurely terminated over the CBF1 transcription unit. All our qPCR oligos are strand-specific and consist of unique sequences in the Arabidopsis genome. In our TSS analysis we could not detect any other peak than the main TSS at the promoter of CBF1. Therefore, fragmentation of the nascent RNA is not necessary and it is highly unlikely that we amplify any other RNA species than the ones originating from the CBF1 promoter (Fig. 5c). These new results showed an increase of RNAPII attached RNA in the 3'-end of the CBF1 exon after 8 hours of cold treatment compared to 4 hours. The further downstream probe in the 3'-UTR of CBF1 showed a decrease of nascent RNA after 8 hours, suggesting that many transcript events do not reach the 3'-UTR of CBF1 after 8 hours. Our combined approaches detect a discrepancy between steady-state levels of RNA and nascent RNA comparing 4h and 8h of cold. We detect increased nascent transcription at the 5'-end of CBF1 at 8h cold that does not result in more CBF1 mRNA. These results support the existence of RNAPII complexes containing nascent RNA that are slowed down/stalled at the 3'-end of the CBF1 exon.

To further address this comment we performed RNAPII ChIP (Fig. 5). Our new results strengthen the conclusions derived from nascent RNA analysis. The ChIP experiments were done with an antibody that recognizes the total RNAPII pool. The relatively small size of the CBF1 gene (~1 kb) limits the experimental design. Nevertheless, we could design qChIP probes for the promoter of CBF1 and the CBF1 transcription unit as well as the proximal promoter and transcription unit of SVK (Fig 5a). We detected an increase of RNAPII occupancy over the CBF1 transcription unit after 8 hours compared to 4 hours of cold treatment. This was in contrast to the CBF1 promoter where RNAPII occupancy decreased after 8 hours. It is also in contrast to the level of CBF1 mRNA at 4 h and 8 h, suggesting that RNAPII complexes are stalled/slowed down over the CBF1 transcription unit after 8 hours of cold treatment.

Moreover, we have included RNAPII ChIP of the *uns-1* mutant after 4 and 8 hours of cold treatment for the CBF1 transcription unit (Fig. 5g). The increase of RNAPII occupancy in the CBF1-body that we saw in wild type was reduced in the *uns-1* mutant. These data suggest that the localized increase of RNAPII (i.e. potential stalling site) was, at least partly, due to the presence of the SVK-asCBF1 cascade. These data suggest that SVK-asCBF1 transcription results in a pile-up of RNAPII in the CBF1 transcription unit. This finding is consistent with RNAPII collision triggered by converging CBF1 and asCBF1 transcription.

All in all, the new data strengthens our hypothesis that RNAPII stalls as a result of RNAPII collision over the CBF1 gene body in response to SVK transcription. The RNAPII stalling site in the CBF1 transcript could reflect a zone for RNAPII collision between RNAPII transcribing CBF1 sense and antisense strands as RNAPII accumulation is reduced when transcription from the SVK-asCBF1 cascade can't invade (i.e. *uns-1* mutant). RNAPII collision likely results in

increased transcriptional termination, which can explain why RNAPII Initiating CBF1 transcription at 8h cold does not generate as much CBF1 mRNA as much as it does at 4h.

Minor points:

I could not find description about depth of the initial TSS-seq analysis, which is better to be included in the article.

A more thorough description of the TSS-seq analysis has been added as a supplementary table (Table S2). Additional details about the TSS-seq analysis have been added to the text (row 73-75 in the manuscript file with highlighted changes), methods and can also be found in Nielsen et al., BioXriv, 2018, doi: <https://doi.org/10.1101/279414>. The method has been described previously (Pelechano et al., Nature Protocols, Volume 11, Issue 2, Pages 359-376, 2016). This reference has been added to the text. Briefly, adjacent TSS separated by not more than 20 bp were merged into TSS clusters. The TSS clusters were annotated by intersection with various genomic features which were extracted from the TxDb.Athaliana.BioMart.plantsmart28 package. We used following definitions: proximal [TSS-500bp, TSS-100bp] and promoters [TSS-100bp, TSS+100bp]. In addition, TSS were further annotated by genomic location (according to TAIR10 and Araport11): promoter, proximal, 5'UTR, exon, intron, 3'UTR, antisense, or intergenic. In case of conflicting annotations, a single annotation was chosen according to the following hierarchy: intergenic < antisense < intron < exon < 5'UTR < proximal < promoter.

Sequences of 3' part of CBF1 transcripts, AS-CBF1, and 5' part of SVALKA should be shown in a supplemental figure to show their positional relationship in the finest resolution.

An additional supplementary figure has been added to visualize the CBF1-SVK region better (Fig. S2).

Reviewer #4:

The work by Kindgren et al. identified a cold induced lncRNA, named as SVALKA, which affects the expression of CBF1 and Arabidopsis freezing tolerance. SVALKA could produce a cryptic lncRNA, asCBF1, which is antisense to CBF1 and represses CBF1 expression. Although the study seems to identify a novel functional lncRNA in plant cold tolerance response, and may also represents a group of lncRNAs with similar functions in responding to environmental stimuli, the authors did not provide necessary evidence to support their conclusions.

We appreciate the call for additional evidence. We provided additional data in the revision that support our conclusions more convincingly.

Major concerns:

1. In the luciferase assay, the authors cloned the +643 bp to +3410 bp upstream of SVALKA to the luciferase. Such a region is long that may contain regulatory regions like enhancers for other genes. The authors argue that the results of uns-1 mutant could prove the direct regulation of SVALKA on CBF1, yet they didn't described how uns-1 was designed and constructed, thus it is impractical to evaluate the reliability of the authors claim.

We would like to clarify that we cloned the whole transcription unit of SVK including the promoter in the LUC-SVK construct, essentially a translational fusion between CBF1 and LUC. LUC expression is hence driven by the CBF1 promoter. The experiment shows that the 3'-UTR including SVK is important for repression of CBF1 (i.e. LUC) during the cold, as compared to the T_{NOS} control less LUC activity is observed.

We agree that the T-DNA lines used in the manuscript need a better explanation. Therefore, we have added a supplementary figure (Fig. S5) describing the T-DNA that has been integrated into the mutants. The positions of the insertions and the transcript boundaries have also been improved (Fig. S2).

2. Could the introduction of oligos antisense to SVALKA revert CBF1 repression and electrolyte leakage under cold treatment?

We thank Reviewer #4 for this suggestion. While oligos are powerful tools to dissect lncRNA function in some systems the delivery into Arabidopsis seedlings is not possible. Even then, oligos are mainly employed to test RNA functions of lncRNA and thus to probe trans-acting functions. We realized that we failed to state more clearly that the uns-1 mutant shows that SVK-asCBF1 acts in cis. SVK is still expressed in the uns-1 mutant expressed but is due to the T-DNA inserted now situated about 4-5 kb away from CBF1 thereby eliminating any cis-action. CBF1 expression is still affected in the uns-1 mutant (Fig 2). These results strongly suggest that SVK has a cis-regulatory function to fine-tune CBF1 by asCBF1, rather than acting as trans-acting lncRNA whose function could be modulated by an oligo-based approach.

3. From the evolutionary point of view, it is hard to understand why Arabidopsis thaliana maintains a mechanism that reduces plant cold resistant ability by producing asCBF1 after long time exposure to cold, when the plant needs stronger cold resistant ability to survive.

As is often observed in biological systems, balanced activity is important. It is clear that CBF1 expression requires tight regulation since overexpression leads to reduced plant fitness. To make this point clearer, we have added a new reference to the text that shows the reduced fitness to CBF1 overexpression (Gilmour et al., Plant Molecular Biology, Issue 54, Page 767-781, 2004). We

find that SVK is important to fine-tune the level of full length CBF1 mRNA, thus it is part of a bigger system to achieve tight regulation. We have also explained the need for tight CBF regulation better in the text (row 47-52 in the manuscript file with highlighted changes).

4. SVALKA is an lncRNA, so the polyA signal shouldn't be important for SVALKA. Why does asCBF1 need to be quickly decayed in WT plants?

We suggest that asCBF1 is rapidly degraded because it is a product of RNAPII collision termination (Prescott and Proudfoot, PNAS, Volume 99, Issue 13, 2002). It thereby expose a free 3'-end that is quickly recognized by the nuclear exosome and degraded. The relative instability of asCBF1 compared to SVK may thus be linked to the transcriptional termination pathway acting on these transcripts. Similarly, the choice of RNAPII transcription termination pathway determines the stability of budding yeast lncRNA species (Marquardt et al., Transcription, Volume 2, Page 145-154, 2011).

Minor points:

1. The color legends of Fig 1a don't match colors used in the figure.

This has been corrected.

2. In the discussion, the authors propose that lncRNA cascades equivalent to SVK-asCBF1 may exist in other plant species. Such argument is better supported by the expression of lncRNAs in the CBF1 neighboring region in other plant species.

We have tried to find appropriate RNA-seq data from different plant species to identify SVK but it has proven more difficult than we initially thought. Clusters of CBF genes can be identified in contigs of genome shotgun sequences. Some RNA-seq data was also available, including of cold-treated samples. However, re-mapping the RNA-seq data to the scaffolds was challenging. Moreover, CBF1 homology can be identified in several clusters due to gene duplications. It remains unclear which region represents the functional homologous cluster. These experiments have to our knowledge not conclusively been reported in species related to Arabidopsis such as cabbage or broccoli. Addressing this experimentally in these different species is beyond the scope of our study. In the revised text, we have therefore downplayed our speculation about SVK conservation.

3. Supplementary Figure 2A should be replaced by a higher quality experimental result.

We wanted to make clear that we do not detect any larger transcripts in the asCBF1 blot (i.e. stable or unstable read-through transcripts from SVK). We have therefore included a higher quality blot with loading control in the supplementary figure (in the revised manuscript Fig S8a).

Reviewers' Comments:

Reviewer #1:

Remarks to the Author:

I think authors address my concern.

Reviewer #2:

Remarks to the Author:

The manuscript has been greatly improved and I now recommend publication.

There remain some minor points:

Line 36, .."can by itself can" second can needs to be removed.

Line 102-108: While we appreciate that the RT analysis from the CBF1:TNOS construct description was taken in in response to reviewer 1, it does not logically link with the description of the previous experiment.

Line 110-117 it is nowhere described how the T-DNA insert lines were identified and screened for.

Line 121-126 the description of the cis versus trans effect of SVK is still misleading: "The combination of high level of SVK expression and increased CBF1 expression in uns-1 argues against a trans-acting function for the SVK lncRNA (Fig. 2d-e). In this mutant, SVK is expressed 4-5 kb away from CBF1, but CBF1 is still affected."

What does affected mean in this context? Is it affected compared to wildtype? i.e. is it expressed at higher levels than wild type? Or is it still regulated, i.e. is its expression still controlled by SVK?

The data suggests that CBF1 in uns-1 is expressed at higher maximal levels, but with a similar kinetic and dynamic-change (not range though). This argues that in uns-1 plants CBF1 is not a strongly suppressed by SVK as in wildtype, but that an additional level of control, which reduces CBF1 levels at later time points is still functioning. The maintenance of this later control cannot be explained by the disruption of a cis-acting mechanism.

Line 137: ° C should be added to the temperatures.

Line 140: has instead of have.

Line 167: "...exosome target with expression..(levels) correlated to SVK" I suggest to insert 'levels' to facilitate understanding of this sentence.

Line 190: To me the xrn3-3 experiment description is not clear. In Xrn3-3 plants one would expect stabilisation of the downstream 3' RNA product still attached to the polymerase. However, in order to be a substrate to Xrn3, the transcript has to be cleaved somewhere and is therefore not expected to be continuous with SVK. In other words, SVK northern probe should not be able to detect RNA accumulated in xrn3-3.

However, apart from these minor points (that I could also have misunderstood) I feel that the manuscript is now fully understandable to anyone within and outside the plant community.

Reviewer #3:

Remarks to the Author:

Reviewer 3

By preparation of new Figure 5 (especially panels e, f and g), the authors have succeeded to suggest a model of transcriptional interference by the antisense transcription. I now recommend this article for publication on Nature Communications.

Minor points

Figure 1a Colors of bar graph do not match with ones in explanation (Promoter, Proximal, 5'UTR, Exon....).

Figure 1b Vertical axis: Fold change is better than Relative signal intensity.

Figure 2d & e Colors of svk-1 and uns-1 in bar graphs do not match with the one in explanations.

Figure 3a Colors of svi-1 and uns-1 in the bar graph do not match with the one in explanation.

Figure 3b Important data.

Figure 4c & d Absolute levels of asCBF1 expression should be shown. If uns-1 and svk-1 do not have any expression of asCBF1 at all, showing expression as fold change is not appropriate.

Figure 5 Important data

Panels c & d Absolute levels of gene expression should be shown.

Comments: The article shows SVK1 modulates Arabidopsis frost tolerance toward weaker side at the range between -5 to -11 oC. I am wondering what is the contribution of SVK1 on induction kinetics of the cold resistance. And, what is not still clear now is biological importance of this regulation. I am looking forward to seeing the next studies.

Reviewer #4:

Remarks to the Author:

The authors have addressed most of my questions, I have no further concerns.

Manuscript NCOMMS-18-11218A
Response to reviewers

Reviewer's comments are marked in **blue** with our responses in **bold black**.

Reviewer #1 (Remarks to the Author):

I think authors address my concern.

We thank reviewer 1 for the suggestions that improved our manuscript.

Reviewer #2 (Remarks to the Author):

The manuscript has been greatly improved and I now recommend publication.

There remain some minor points:

Line 36, .."can by itself can" second can needs to be removed.

Done

Line 102-108: While we appreciate that the RT analysis from the CBF1:TNOS construct description was taken in in response to reviewer 1, it does not logically link with the description of the previous experiment.

In the revised manuscript, we have tried to make the logic more clear.

Line 110-117 it is nowhere described how the T-DNA insert lines were identified and screened for.

We have added this information to the method section (Lines 324-326 in the manuscript file with highlighted changes).

Line 121-126 the description of the cis versus trans effect of SVK is still misleading: "The combination of high level of SVK expression and increased CBF1 expression in uns-1 argues against a trans-acting function for the SVK lncRNA (Fig. 2d-e). In this mutant, SVK is expressed 4-5 kb away from CBF1, but CBF1 is still affected."

What does affected mean in this context? Is it affected compared to wildtype? i.e. is it expressed at higher levels than wild type? Or is it still regulated, i.e. is its expression still controlled by SVK?

We have tried to make this clearer in the text (Lines 118-120 in the manuscript file with highlighted changes).

The data suggests that CBF1 in uns-1 is expressed at higher maximal levels, but with a similar kinetic and dynamic-change (not range though). This argues that in uns-1 plants CBF1 is not a strongly suppressed by SVK as in wildtype, but that an additional level of control, which reduces CBF1 levels at later time points is still functioning. The maintenance of this later control cannot be explained by the disruption of a cis-acting

mechanism.

We do not claim that the SVK mechanism is acting alone in regulating CBF1 but has more of a fine-tuning role. All regulatory roles at the CBF1 promoter are still present in the uns-1 mutant.

Line 137: ° C should be added to the temperatures.

Done

Line 140: has instead of have.

Done

Line 167: "...exosome target with expression..(levels) correlated to SVK" I suggest to insert 'levels' to facilitate understanding of this sentence.

Done

Line 190: To me the *xrn3-3* experiment description is not clear. In *Xrn3-3* plants one would expect stabilisation of the downstream 3' RNA product still attached to the polymerase. However, in order to be a substrate to *Xrn3*, the transcript has to be cleaved somewhere and is therefore not expected to be continuous with SVK. In other words, SVK northern probe should not be able to detect RNA accumulated in *xrn3-3*.

The reviewer is correct. We used the *xrn3-3* mutant to show that CBF1 and SVK expression are similar to wild-type. If the SVK read-through would be a *XRN3* target, we would expect a more stable asCBF1 transcript and that we would be able to detect asCBF1 in the *xrn3-3* mutant. We have therefore added "read-through" to the text to make this clear (Line 189 in the manuscript file with highlighted changes).

However, apart from these minor points (that I could also have misunderstood) I feel that the manuscript is now fully understandable to anyone within and outside the plant community.

We thank reviewer 2 for the suggestions that improved our manuscript.

Reviewer #3 (Remarks to the Author):

By preparation of new Figure 5 (especially panels e, f and g), the authors have succeeded to suggest a model of transcriptional interference by the antisense transcription. I now recommend this article for publication on Nature Communications.

Minor points

Figure 1a Colors of bar graph do not match with ones in explanation (Promoter, Proximal, 5'UTR, Exon....).

Colors have been changed.

Figure 1b Vertical axis: Fold change is better than Relative signal intensity.

Axis title has been changed.

Figure 2d & e Colors of svk-1 and uns-1 in bar graphs do not match with the one in explanations.

Colors have been changed.

Figure 3a Colors of svi-1 and uns-1 in the bar graph do not match with the one in explanation.

Colors have been changed.

Figure 3b Important data.

Agreed.

Figure 4c & d Absolute levels of asCBF1 expression should be shown. If uns-1 and svk-1 do not have any expression of asCBF1 at all, showing expression as fold change is not appropriate. In panel 4c, we could detect asCBF1 expression in both hen2-2uns-1 and hen2-2svk-1 mutants. We therefore believe that our analysis is correct. In panel 4d, the transcription levels of SVK read-through were not detected in steady-state RNA or the Mock-IP sample (i.e. they did not have a Ct value of below 40 in our qPCR analysis). We have added the Ct values for our reference genes in the Source Data file. These show that we can detect the reference in all samples confirming that our RT reaction was successful.

Figure 5 Important data. Panels c & d Absolute levels of gene expression should be shown. See our response above for 4d. Not detected means that the level of expression was below the detection limit of our qPCR analysis. We have added the Ct values for our reference genes in the Source Data file. These show that we can detect the reference in all samples confirming that our RT reaction was successful.

Comments: The article shows SVK1 modulates Arabidopsis frost tolerance toward weaker side at the range between -5 to -11 oC. I am wondering what is the contribution of SVK1 on induction kinetics of the cold resistance. And, what is not still clear now is biological importance of this regulation. I am looking forward to seeing the next studies.

We agree with the reviewer. This will be an interesting line of research to follow up in the future. It was out of the scope of this manuscript, however.

Reviewer #4 (Remarks to the Author):

The authors have addressed most of my questions, I have no further concerns.

We thank reviewer 4 for the suggestions that improved our manuscript.